# The Quasimesenchymal Pancreatic Ductal Epithelial Cell Line PANC-1—A Useful Model to Study Clonal Heterogeneity and EMT Subtype Shifting

**DOI:** 10.3390/cancers14092057

**Published:** 2022-04-19

**Authors:** Hendrik Ungefroren, Isabel Thürling, Benedikt Färber, Tanja Kowalke, Tanja Fischer, Leonardo Vinícius Monteiro De Assis, Rüdiger Braun, Darko Castven, Henrik Oster, Björn Konukiewitz, Ulrich Friedrich Wellner, Hendrik Lehnert, Jens-Uwe Marquardt

**Affiliations:** 1First Department of Medicine, University Hospital Schleswig-Holstein, Campus Lübeck, Center of Brain, Behavior and Metabolism (CBBM), University of Lübeck, D-23538 Lübeck, Germany; isabel.thuerling@student.uni-luebeck.de (I.T.); tanja.kowalke@student.uni-luebeck.de (T.K.); tanja.fischer@student.uni-luebeck.de (T.F.); darko.castven@uksh.de (D.C.); jens.marquardt@uksh.de (J.-U.M.); 2Clinic for Surgery, University Hospital Schleswig-Holstein, Campus Lübeck, University of Lübeck, D-23538 Lübeck, Germany; benedikt.faerber@student.uni-luebeck.de (B.F.); ruediger.braun@uksh.de (R.B.); ulrich.wellner@uksh.de (U.F.W.); 3Institute of Pathology, University Hospital Schleswig-Holstein, Campus Kiel, D-24105 Kiel, Germany; bjoern.konukiewitz@uksh.de; 4Institute for Neurobiology, Center of Brain, Behavior and Metabolism (CBBM), University of Lübeck, D-23538 Lübeck, Germany; leonardo.deassis@uni-luebeck.de (L.V.M.D.A.); henrik.oster@uksh.de (H.O.); 5University of Salzburg, A-5020 Salzburg, Austria; hendrik.lehnert@sbg.ac.at

**Keywords:** pancreatic ductal adenocarcinoma, epithelial–mesenchymal plasticity, EMT, MET, proinflammatory cytokines, transforming growth factor β, transdifferentiation, PANC-1, phenotype, intratumoral heterogeneity

## Abstract

**Simple Summary:**

Malignant tumors often escape therapy due to clonal propagation of a subfraction of drug-resistant cancer cells. The underlying phenomenon of intratumoral heterogeneity is driven by epithelial–mesenchymal plasticity (EMP) involving the developmental programs, epithelial–mesenchymal transition (EMT), in which epithelial cells are converted to invasive mesenchymal cells, and the reverse process, mesenchymal–epithelial transition (MET), which allows for metastatic outgrowth at distant sites. For therapeutic targeting of EMP, a better understanding of this process is required; however, cellular models with which to study EMP in pancreatic ductal adenocarcinoma (PDAC) are scarce. Using single-cell clonal analysis, we have found the PDAC cell line, PANC-1, to consist of cells with different E/M phenotypes and functional attributes. Parental PANC-1 cultures could be induced in vitro to shift towards either a more mesenchymal or a more epithelial phenotype, and this bidirectional shift was controlled by the small GTPases RAC1 and RAC1b, together identifying PANC-1 cells as a useful model with which to study EMP.

**Abstract:**

Intratumoral heterogeneity (ITH) is an intrinsic feature of malignant tumors that eventually allows a subfraction of resistant cancer cells to clonally evolve and cause therapy failure or relapse. ITH, cellular plasticity and tumor progression are driven by epithelial–mesenchymal transition (EMT) and the reverse process, MET. During these developmental programs, epithelial (E) cells are successively converted to invasive mesenchymal (M) cells, or back to E cells, by passing through a series of intermediate E/M states, a phenomenon termed E–M plasticity (EMP). The induction of MET has clinical potential as it can block the initial EMT stages that favor tumor cell dissemination, while its inhibition can curb metastatic outgrowth at distant sites. In pancreatic ductal adenocarcinoma (PDAC), cellular models with which to study EMP or MET induction are scarce. Here, we have generated single cell-derived clonal cultures of the quasimesenchymal PDAC-derived cell line, PANC-1, and found that these differ strongly with respect to cell morphology and EMT marker expression, allowing for their tentative classification as E, E/M or M. Interestingly, the different EMT phenotypes were found to segregate with differences in tumorigenic potential in vitro, as measured by colony forming and invasive activities, and in circadian clock function. Moreover, the individual clones the phenotypes of which remained stable upon prolonged culture also responded differently to treatment with transforming growth factor (TGF)β1 in regard to regulation of growth and individual TGFβ target genes, and to culture conditions that favour ductal-to-endocrine transdifferentiation as a more direct measure for cellular plasticity. Of note, stimulation with TGFβ1 induced a shift in parental PANC-1 cultures towards a more extreme M and invasive phenotype, while exposing the cells to a combination of the proinflammatory cytokines IFNγ, IL1β and TNFα (IIT) elicited a shift towards a more E and less invasive phenotype resembling a MET-like process. Finally, we show that the actions of TGFβ1 and IIT both converge on regulating the ratio of the small GTPase RAC1 and its splice isoform, RAC1b. Our data provide strong evidence for dynamic EMT–MET transitions and qualify this cell line as a useful model with which to study EMP.

## 1. Introduction

Variability across tumors of different patients or across clusters of malignant cells within the same patient is known as intertumoral heterogeneity, while intratumoral heterogeneity (ITH) refers to variability within an individual tumor in a patient. ITH at the cellular level reflects a high degree of plasticity of cancer cells as a result of repeated accumulation of genetic, epigenetic and/or metabolic alterations. Cancer cell plasticity allows for continuous and reversible adaptation to changing conditions orchestrated by signals from the tumor microenvironment [1]. It helps the tumor to escape selection pressures, i.e., anticancer drug treatment, by allowing a subfraction of resistant cells to clonally evolve, eventually resulting in therapy failure or relapse [2,3]. Understanding the cellular and molecular alterations required for the establishment of ITH and cellular plasticity may provide opportunities to exploit these processes for combating metastasis and drug resistance.

Cellular plasticity involves the reactivation of epithelial–mesenchymal transition (EMT), a developmental program that results in the dynamic interconversion of sessile epithelial (E) to motile mesenchymal (M) cells. EMT is considered an early event in the development of some tumors, i.e., in pancreatic ductal adenocarcinoma (PDAC) [4], and, in addition to invasion and metastasis, is associated with growth inhibition, enhanced patient survival, genomic instability, metabolic reprogramming, drug resistance and immune suppression [1,5,6,7,8]. This is in part due to the generation of cancer stem cells (CSCs), dormant and slow cycling cells with self-renewal potential and enhanced drug efflux properties, which maintain cancer cell survival and growth upon treatment [9]. The EMT, which is induced by stromal factors, hypoxia, growth factors or inflammatory cytokines, is inherently reversible, with the reverse process termed M–E transition (MET). Although MET is increasingly recognized as the principal mechanism for establishing metastasis following distant colonization of circulating tumor cells, it has received much less attention than EMT. MET involves the re-expression of E cell markers, such as E-cadherin (ECAD) or cytokeratin 19 (CK19), and an accompanying inhibition of M markers, such as vimentin (VIM), RAC1 or SNAIL [10]. The current paradigm states that EMT drives carcinoma cells to disseminate from the primary tumor, while MET drives disseminated carcinoma cells to reinitiate proliferative programs necessary for the formation of secondary tumors [10]. The ability of disseminated tumor cells to reestablish epithelial phenotypes via MET programs may in fact represent the rate-limiting step in the metastatic cascade [10].

EMT and MET have long been viewed as binary switches between E or M cell populations. More recently, however, the term E–M plasticity (EMP) is favored as compared to EMT–MET [6,11] to account for the bidirectional dynamics of EMT and MET phenotypes, wherein the cancer cells can reside in all three EMP states, E (well differentiated), M (poorly differentiated, stem-like) and hybrid E/M. The E/M state is generated by sequential acquisition of M traits but partial retainment of their previously expressed E traits [12,13]. This intrinsic mechanism of bidirectional interconversions between E and M phenotypes [14,15,16,17] and stem and non-stem compartments [18] has been reported for different types of cancer and represents the driving force of ITH.

There is currently a lack of cellular models with which to analyse EMP. It would, therefore, be useful to have access to an experimental system that allows for EMT or MET induction by shifting cells from E to M phenotypes or vice versa. Previous studies from our group have shown that the propensity of permanent PDAC-derived cell lines to transdifferentiate into pancreatic endocrine precursors or pancreatic β cell-like cells depends on their histological or phenotypic EMT subtype. Specifically, the quasimesenchymal (QM) cell lines PANC-1 and MIA PaCa-2 exhibit the highest potential for activating a ductal-to-endocrine transdifferentiation transcriptional program (deTDtP) [19]. The deTDtP is controlled in an antagonistic fashion by the small GTPase RAC1 and its negative regulator and *RAC1* splice isoform, RAC1b, through regulation of pluripotency genes, possibly reflecting the presence of CSCs with high deTD potential [20].

In accordance with its QM nature, PANC-1 cells have been classified as cells with complete EMT (cEMT), as evidenced by low or absent protein expression of ECAD and transcriptional regulation of its gene (*CDH1*) as opposed to E/classical PDAC cells, i.e., BxPC-3, with partial EMT (pEMT), high ECAD protein and its regulation at the level of subcellular localization [21]. However, within the cEMT group of PDAC cell lines, PANC-1 is more “E-like”, as evidenced by readily detectable protein expression of ECAD and RAC1b [22] in contrast to MIA PaCa-2, which possesses an extreme M phenotype [21] and hence completely lacks ECAD and RAC1b expression [22]. By microscopic inspection, parental cultures of PANC-1 have been described as pleomorphic, consisting of cells with different morphological patterns and variable expression of E, M, neuroendocrine and stem cell markers [23]. However, a rigorous characterization and functional analysis of these phenotypically dissimilar cells, e.g., after generation and propagation of single cell-derived clones—a strategy employed successfully for other tumor types [14]—has not yet been performed. Preliminary evidence for clonal variation in cellular plasticity recently came from our latest study with single cell-derived clones of PANC-1 depleted of RAC1b by genomic editing, which exhibit strong variations in expression levels of pluripotency factors [20]. This strongly suggested to us that clonal heterogeneity may also exist in non-genetically altered/wild-type cultures of PANC-1 with respect to the simultaneous presence of various EMT phenotypes and possibly the existence of stochastic or induced transitions between them. Here, we implemented single cell clonal propagation to characterize molecular and cellular features of intrinsic heterogeneity with respect to functional attributes of invasion, proliferation, transdifferentiation potential, circadian rhythmicity and specific gene responses. In addition, we were able to identify agents capable of shifting parental PANC-1 cultures towards a more M- or a more E-like phenotype, suggesting the activation of EMT- and MET-like programs, respectively.

## 2. Results

### 2.1. Single Cell-Derived Clones of PANC-1 Reveal Heterogeneity in Morphology and EMT Marker Expression

Single cell-derived clones were derived from PANC-1 and BxPC-3 cells by limited dilution and six and five clones, respectively, of each cell line, analysed further by microscopy and EMT marker expression using immunoblotting. Intriguingly, single cell-derived clones of PANC-1 but not BxPC-3 cells (not shown) presented with dramatic differences in cell morphology and growth patterns (Appendix A). While cultures of P4B9 and P1C3 cells grew in a scattered fashion and contained many spindle-shaped cells, P4B11 and P3D2 grew in clusters with many cobblestone-like cells (Appendix A); clone P3D10 presented with few spindle-shaped and mostly cuboid cells, while clone P1G7 cells were either spindle-shaped or rounded up and refractive, suggestive of senescent cells (Appendix A).

Next, we evaluated the expression of ECAD, VIM and RAC1 in immunoblots. Both ECAD and VIM, but not RAC1, markedly differed in abundance among clones (Figure 1A). Upon signal quantification, we noted that the clones exhibited different ECAD:VIM ratios, which were high for P3D2 and P4B11, low for P1C3 and P4B9, and intermediate for P3D10, P1G7 and the parental cells (Figure 1A). In another series of immunoblot experiments we quantified RAC1 and RAC1b protein levels and calculated their ratio of expression. We observed that the RAC1b/RAC1 ratio largely corresponded to the ECAD:VIM ratio in that it was high in clones P3D2, P4B11 (and P1G7), but lower in P1C3, P4B9 (and P3D10) (Appendix A). Interestingly, when we monitored expression of ECAD in five individual clones of BxPC-3 (B3F8, B1C5, B4D6, B2G7, B1D9) along with the parental cells, no visible differences were evident (Figure 1B). Due to their E nature, VIM protein was undetectable in these cells (Figure 1B).

We then wanted to know whether the single cell-derived PANC-1 clones display a stable phenotype if cultured for longer periods of time or if they eventually re-establish a mixed E/M population similar to parental cultures. To address this issue, we generated, again by limited dilution, single cell-derived subclones of the clones P3D2, P3D10 and P4B11, and compared six of these with respect to morphology (P3D2, P4B11) and expression of ECAD and VIM (P3D2, P3D10). For all three clones, both the cell morphology/growth pattern (Appendix A) as well as the protein levels of ECAD and VIM (Figure 1C) were quite homogenous among the subclones as evidenced by very similar ECAD:VIM ratios (Figure 1C). Together, this indicated that the phenotype of the various clones is stable under standard culture conditions and that they apparently do not re-establish a mixed E/M population similar to parental cultures.

PANC-1 cells were reported to possess neuroendocrine differentiation and to express the neuroendocrine marker synaptophysin (SYP) [23]. Again, the six PANC-1 clones from above exhibited clear differences in SYP expression as determined by qPCR analysis (Appendix A).

Taken together, we confirmed by analysis of individual clones the ITH observed previously in parental PANC-1 cultures by microscopic and immunocytochemical analyses. Based on cell morphology and EMT marker expression, we were able to tentatively classify P1C3 and P4B9 as M, P3D2 and P4B11 as E, and P3D10 and P1G7 as E/M. Clone P1G7 was exceptional in that its cells presented with a senescence-like phenotype and high abundance of VIM protein, in agreement with the earlier identification of VIM as a marker of cellular senescence [24,25].

### 2.2. Single Cell-Derived Clones of PANC-1 Reveal Heterogeneity in Tumorigenic Potential with Respect to Stemness and Migratory Activity

The malignant behavior of a tumor is driven by CSCs that foster tumor growth and metastatic dissemination. To screen for potential heterogeneity in invasive potential, we analysed the six individual PANC-1 clones mentioned above by xCELLigence-based real-time cell migration assay and found them to differ dramatically in basal migratory activity (Figure 2A). Of note, the clones with the highest migratory activity (P1C3, P3D10, P4B9) exhibited the lowest ECAD:VIM ratios, while those of the clones with weak migratory capacity (P4B11, P1G7, P3D2) were higher (Appendix A). Likewise, clones with high migration potential also exhibited a lower RAC1b/RAC1 ratio than clones with low migratory activity (Appendix A), in accordance with earlier data from breast cancer cell lines [26]. In addition, we observed that clone P1C3 was SYP high, consistent with the association of neuroendocrine differentiation with an aggressive behavior in certain cancers, i.e., the prostate [27]. In contrast, four individual clones of BxPC-3 (B4D6, B2G7, B3F8, B1D9) were much more uniform in migratory activity (Figure 2B). The migratory behavior of parental PANC-1 and BxPC-3 cells in relation to the clones is shown in Appendix A, respectively.

Next, we assessed the stemness potential of two PANC-1 clones, namely, P1C3 (exhibiting the highest invasive activity) and P4B11 (presenting with low invasive activity), by assessing their ability to form clonogenic colonies in 2-D. To this end, in this colony formation assay (CFA), P1C3 cells generated 8.4-fold more colonies than P4B11 cells (Figure 2C). These data show that single cell-derived clones of PANC-1, but not BxPC-3, present with strong variations in cellular phenotypes associated with invasion and metastasis and that the two PANC-1 clones classified as M (P1C3, P4B9) exhibit a more malignant phenotype in vitro than the E-like clones, P3D2 and P4B11.

### 2.3. Single Cell-Derived Clones of PANC-1 Reveal Heterogeneity in TGFβ1-Mediated Regulation of Growth and Individual TGFβ Target Genes

Phenotypic heterogeneity may also extend to the cellular response to growth factor stimulation. To study this in more detail, we evaluated various responses to transforming growth factor (TGF)β given its potent EMT-promoting function. We first quantitatively measured cell viability, reflecting growth regulation, in response to TGFβ1 using a resazurin-based assay. Upon entering living cells, resazurin is reduced to resorufin, a compound that is red in color and highly fluorescent (Figure 3A). Results revealed strong differences in TGFβ1’s cytostatic effect among the set of the six PANC-1 clones. While P1G7, P3D2, P4B9 and parental cells as control responded with growth inhibition, P1C3 and P4B11 were refractory and P3D10 even showed a small stimulatory effect (Figure 3A). To confirm the effects of TGFβ1 on proliferative activities more directly, we additionally performed cell counting assays. The basal/steady-state proliferative activity was quite different among the six clones tested with a 2.2-fold difference between the clones with the highest and the lowest cell numbers, P3D2 and P3D10, respectively (Appendix A). While the parental cells reacted with growth inhibition to TGFβ1 stimulation, in agreement with previous studies [28,29], the clones responded in a way that essentially mirrored the results of the resazurin-based assays; clones P1G7 and P4B9 were growth-inhibited, while clones P1C3, P3D2 and P4B11 were insensitive relative to untreated controls (Appendix A). As in the viability assay, clone P3D10 showed an increase in cell numbers; however, this effect tightly missed statistical significance (Appendix A).

Next, we evaluated in the various PANC-1 clones by qPCR analysis basal and TGFβ1-induced expression of the established TGFβ target gene, *SERPINE1* (encoding plasminogen activator-inhibitor type 1, PAI-1). While basal expression of this gene was fairly similar among clones, its expression in response to TGFβ1 challenge was induced in all clones, except for P3D2, albeit to various extents (Figure 3B). Interestingly, the clones with the lowest ECAD/VIM ratio (P1C3, P3D10, P4B9) exhibited a greater *SERPINE1* response to TGFβ1 treatment than the E-type clones P1G7 and P4B11, as well as another E-type clone (P3D2), for which the TGFβ1 effect was not even statistically significant.

Finally, we monitored by immunoblotting in five clones and in parental cells the TGFβ1 response of SNAIL, a master transcription factor of EMT, with ECAD as control. Again, while basal levels of SNAIL were quite similar, TGFβ1-induced protein levels varied up to a factor of 4 among the clones tested (Figure 3C). Similar results were obtained for the related SLUG (encoded by *SNAI2*). The three clones that were the most responsive to TGFβ1 with induction of SLUG mRNA (P1C3, P3D2, P4B11) (Figure 3D) also exhibited the largest induction in SNAIL protein (see Figure 3C). We conclude that clonal heterogeneity among PANC-1 clones is also evident from the cells’ sensitivity to TGFβ1 with respect to proliferative activity and regulation of target genes involved in EMT and invasion.

### 2.4. Single Cell-Derived Clones of PANC-1 Reveal Heterogeneity in Ductal-to-Endocrine Transdifferentiation Potential and Circadian Clock Function

Having shown previously that (parental) PANC-1 can be stimulated to activate a deTDtP using a cocktail of three proinflammatory cytokines, IFNγ, IL1β and TNFα (IIT) [20,30], we asked whether the cells’ propensity for deTD may vary with the EMT phenotype. To this end, individual clones of PANC-1 cells dramatically differ in their ability to induce expression of Insulin (encoded by *INS*, upper graph) or Neurogenin-3 (encoded by *NEUROG3*, lower graph) following deTD culture (Figure 4A).

PANC-1 cells have been reported to show circadian oscillations—although with small amplitudes and short periods—using Luc-based reporters of the clock genes, *Bmal1* or *Per2* [31]. Since *BMAL1* has been implicated in EMP of colorectal cancer cells [32], we employed *Bmal1::Luc* plasmid transfection into parental or clonal cultures of PANC-1 and measured circadian luminescence rhythms after synchronization with dexamethasone (dexa). We were unable to detect marked *Bma1l::Luc* rhythms in PANC-1 parental cultures after synchronization with dexa, forskolin or serum shock (Figure 4B, upper left panel). As in PANC-1 parental cells, no rhythms but some transient responses in luciferase activity were detectable after dexa treatment in clones P4B11, P2E8, P1C3 and P1G7 (Figure 4B, upper right panels and lower left panel). P3D10, P3D2 and MIA PaCa-2 cells showed rhythmic but highly dampened *Bmal1* luciferase activity after dexa (Figure 4B, lower right panels). While P3D2 and MIA PaCa-2 cells displayed periodicity in the circadian range, P3D10 had a longer period at 33–34 h. Conversely, P3D10 showed smaller dampening rates compared to P3D2 and MIA PaCa-2 (Appendix A). In conclusion, circadian clock function was affected in a clone-dependent manner with the most marked rhythmicity observed in P3D2, P3D10 and MIA PaCa2 cells; however, an association with EMT phenotypes was not possible. In any case, though, rhythms were strongly dampened, suggesting an overall weak clock function.

### 2.5. Exogenous TGFβ1 Induces a More Mesenchymal and Invasive Phenotype in Parental Cultures of PANC-1 Cells

The data presented above have shown that PANC-1 cells exist in different phenotypes among which dynamic shifts may occur either stochastically or induced by external cues, i.e., growth factors/cytokines. Moreover, clonal PANC-1 cultures were all TGFβ-sensitive, as evidenced by at least one cellular response associated with M conversion, such as growth inhibition, downregulation of ECAD, or upregulation of *SERPINE1* and SNAIL. Subsequent experiments designed to identify agents that are able to induce phenotype shifting were therefore performed in parental cultures. In parental PANC-1 cells, treatment with TGFβ1 indeed phenocopied a M shift, as evidenced by a spindle-shaped morphology, scattered growth pattern (Appendix A) and downregulation of the epithelial markers EpCAM, Claudin 7 (CLDN7), ECAD and CK19 (Figure 5A). The silencing of *CDH1* by TGFβ1 at the mRNA level rather than removal of the protein from the cell surface and transport to intracellular stores is consistent with a decrease in ECAD protein levels [22] and attests to the cEMT phenotype of this cell line [21]. Like the single cell-derived clones, parental PANC-1 cells responded to TGFβ1 treatment with transcriptional induction of *SERPINE1* and *SNAI1* (Figure 5B), in agreement with induction of SNAIL and SLUG by this growth factor (Figure 3C,D). As expected from the data in Figure 3D, we observed upregulation of *SNAI2* and *BGN*, the gene encoding the small leucine-rich proteoglycan and M marker, Biglycan (Figure 5B). Moreover, TGFβ1 strongly increased invasive activity of PANC-1 cells, as demonstrated by real-time cell invasion assays using Matrigel as barrier (Figure 5C). Together, the data show that PANC-1 cells, despite their cEMT phenotype [21], can be shifted further towards the M end of the E–M spectrum by activation of TGFβ1 signaling.

### 2.6. A Combination of IFNγ, IL1β and TNFα Induces a More Epithelial and Less Invasive Phenotype

The treatment of PANC-1 cultures with IIT triggered a differentiation process resulting in cells that closely resemble pancreatic progenitors/β cell-like cells, which are considered epithelial in nature. Conversely, β cell dedifferentiation is associated with the induction of an EMT gene signature, including downregulation of OVO Like Zinc Finger 2 (OVOL2), a marker of MET and transcriptional repressor of the EMT TF, Zeb2 [33]. Together, this suggested the exciting possibility that IIT-treated PANC-1 cells have (partially) lost their QM phenotype and undergone a MET-like program. To this end, IIT treatment indeed resulted in a shift in EMT marker expression, with downregulation of VIM, RAC1 (Figure 6A) and SNAIL (Appendix A). Likewise, treatment of another QM-type PDAC cell line with an extreme M phenotype, MIA PaCa-2, with IIT for 72 h reduced the abundance of the VIM and RAC1 proteins in a time-dependent manner (Figure 6B). Extension of treatment to 120 h further decreased VIM and RAC1 levels in PANC-1 (Appendix A) but not in MIA PaCa-2 cells (Figure 6B). Concomitant with downregulation of M markers, we noted augmented expression by IIT of Claudin-4 (CLDN4), an integral constituent of tight junctions and an inhibitor of invasion and metastasis in pancreatic cancer cells [34] (Figure 6C).

To reveal whether the observed changes in protein abundance of M markers were transcriptionally mediated, we performed qPCR analyses in parental PANC-1 and MIA PaCa-2 cells transdifferentiated for 72 h with IIT. We observed a decrease in mRNA levels of *VIM*, *RAC1* and *SNAI1* (Figure 6D) in PANC-1, and *VIM* and *RAC1* in MIA PaCa-2 cells (Figure 6D). In contrast, monitoring of E markers revealed upregulation of *CK19* (Figure 6D) but not *CDH1* (not shown) in PANC-1 and upregulation of *CDH1* but not *CK19* in MIA PaCa-2 cells (Figure 6D). We further reasoned that IIT-mediated E conversion should also induce or activate specific TFs that drive the E phenotype, i.e., Grainyhead-like 2 (GRHL2) or OVOL2, which also serve as biomarkers for MET and stabilize hybrid E/M phenotypes [35]. Of note, both *GRHL2* and *OVOL2* were strongly upregulated by IIT in PANC-1 and MIA PaCa-2 cells relative to controls (Figure 6D).

The downregulation of invasion promoters (RAC1, VIM, SNAIL) and concurrent upregulation of invasion suppressors (ECAD, CLDN4) in IIT-treated PANC-1 and/or MIA PaCa-2 cells may have functional consequences. Specifically, we reasoned that a MET-like process should render the cells less malignant. To this end, PANC-1 cells pre-treated for 48 h with IIT exhibited less invasive activity (Figure 6E). From the data presented in this section, we conclude that as part of an IIT-induced differentiation process, M proteins including master EMT TFs are downregulated, while E proteins including master MET TFs are upregulated, ultimately leading to the acquisition of a more E-like and hence more benign phenotype.

### 2.7. TGFβ1 and IIT Differentially Regulate RAC1b—TGFβ1 and TII Converge on Shifting the Ratio of RAC1 to RAC1b

Alternative mRNA splicing is increasingly recognized as an integral mechanism in the regulation of EMT and MET programs [36,37]. A major role in regulating mRNA splicing is played by TGFβ signaling through differential regulation of paired splice variants of which one promotes an E phenotype (or MET) and the other an M phenotype (or EMT) [38]. Such a pair of splice variants is represented by RAC1 and its splice isoform, RAC1b, with RAC1 promoting the M and RAC1b the E state [22]. Given the strong impact of RAC1b and RAC1 on TGFβ1-induced EMT [39] and the above finding of downregulation of RAC1 and upregulation of RAC1b by IIT treatment, we considered the possibility that RAC1 and/or RAC1b, too, are subject to regulation by TGFβ1 and/or IIT. To validate this assumption, we went on to monitor RAC1b expression in parental PANC-1 cultures after treatment with TGFβ1 for 3 d. Intriguingly, RAC1b protein levels dropped to 36.1 ± 10.2% (*p* < 0.05) of untreated controls, while levels of RAC1 remained unchanged (Figure 7A). Concomitant treatment of cells with the ALK5 inhibitor SB431542 (1 µM), but not the chemically related p38 MAPK inhibitor SB203580 (10 µM), was able to prevent downregulation of RAC1b by TGFβ1. Since different cellular responses to TGFβ1 may require different ligand thresholds, we determined the TGFβ1 concentrations and treatment duration required for maximal suppression of RAC1b protein levels. A dose–response assay in PANC-1 cells revealed that TGFβ1 concentrations of 0.5–5 ng/mL were required to achieve a robust downregulation (Figure 7B).

To determine whether RAC1b downregulation by TGFβ1 is controlled at the post-transcriptional or transcriptional level, we employed PANC-1 cells with stable ectopic expression of a HA-tagged version of RAC1b. These cells have been characterized previously and shown to resist TGFβ1-induced upregulation of cell migration [40]. Of note, the ectopically expressed RAC1b but not endogenous SNAIL protein was refractory to regulation by TGFβ1, hence its abundance remained stable over the 3 or 10 d observation period (Figure 7C). These results indicate that Rac1b protein stability is not affected by TGFβ1, implying regulation at the mRNA level, which was subsequently confirmed by qPCR analysis with primers directed against exon 3b of *RAC1*. Treatment with TGFβ1 for 24 h resulted in downregulation of RAC1b steady-state mRNA levels to 47.5 ± 16.8% of controls (*p* = 0.033, *n* = 3) (Figure 7D). Consistent with the absence of a regulatory effect of TGFβ1 on protein abundance of RAC1 (Figure 7A,B), no significant changes in RAC1 mRNA levels were noted (Figure 7D).

Finally, given the association of RAC1b with the E phenotype, we reasoned that RAC1b expression should increase during the MET process. Intriguingly, after stimulation of PANC-1 cells with the proinflammatory cytokine mix IIT, protein levels of RAC1b were elevated (Figure 7E), while those of RAC1 were suppressed (Figure 7E), as already demonstrated in Figure 6A. From these data we conclude that TGFβ1 and the proinflammatory cytokine cocktail differentially target RAC1b, with RAC1b being inhibited at the level of transcription or RNA splicing rather than protein stability. Our data further imply that down- or upregulation of endogenous RAC1b protein by TGFβ1 or IIT is instrumental in increasing the M or E phenotype, respectively.

## 3. Discussion

The characterization of multiple subpopulations from mammary and skin tumors suggested that tumor cells with hybrid phenotypes are more efficient in dissemination and metastasis [41,42,43]. Other studies have found that cancer cells mostly transit between E/M and hybrid states but rarely undergo complete EMT during metastasis [44], providing direct evidence of EMP under physiological conditions [4,7,45]. However, the molecular drivers and the dynamics of the stochastic or induced state transitions, which allow cancer cells to switch between phenotypic states and form the basis of ITH remain to be defined [46]. Given the current lack of experimental models to analyse EMP, it would be useful to have a cellular model, which allows for reversible bidirectional EMT–MET shifting and analysis of intermediate states. PANC-1 cells appear to be suitable for such a cellular model system based on the following features: (i) a marked cellular heterogeneity in parental cultures [23], possibly reflecting spontaneous, induced or clonal phenotypic variation, (ii) a documented cellular plasticity with respect to deTD [19] and (iii) the presumed presence of CSCs, as evidenced by expression of stem cell/pluripotency markers and their antagonistic control by RAC1 and RAC1b [20]. To study this in greater depth, we first generated from PANC-1 parental cultures by limited dilution single cell-derived clones and characterized these with respect to various cellular responses, such as EMT marker expression, migratory potential, sensitivity to growth factor (TGFβ1) stimulation, deTD potential and circadian rhythmicity. As control, we employed in some assays single cell-derived clones of the epithelial, well-to-moderately differentiated cell line, BxPC-3, in which all cells of the population appear morphologically similar.

The PANC-1-derived clones exhibited dramatic differences with respect to cell morphology and expression of E, M and neuroendocrine markers. Based on the ratio of ECAD:VIM protein levels and invasive behavior in vitro, we delineated subpopulations designated as E (P3D2, P4B11), E/M (P3D10) or M (P1C3, P4B9). Intriguingly, the PANC-1 clone with the lowest ECAD:VIM ratio but highest SYP and SNAIL expression (P1C3) also exhibited the highest migratory activity. Conversely, the clones with the highest ECAD:VIM ratio (P2D3, P4B11) were the least migration-active ones. We also noted variations in the response of PANC-1 clones to TGFβ1 with respect to growth inhibition and target gene expression, e.g., PAI-1, ECAD and SNAIL, and to treatment with proinflammatory cytokines (IIT) regarding transdifferentiation to insulin-expressing cells.

Assessment of circadian clock rhythms in parental PANC-1 cultures and subclones in *Bmal1*::*Luc* assays reveled highly clone-specific effects. In line with previous findings from other cancer cells (reviewed in Ref [47]), a lack of circadian clock function in subclones reflected enhanced migration in clones such as P1C3 and P4B9. On the other hand, no clear correlation between E/M phenotype and circadian clock function can be drawn, suggesting a complex relationship between cellular clock functionality and tumor phenotype [48]. Overall, weak or absent clock rhythmicity supports the idea of chronodisruption as a hallmark of cancer cells [49,50]. In PANC-1 and MIA PaCa-2 cells this may be a consequence of mutant KRAS expression [51].

The clonal functional heterogeneity revealed in Figure 1 may result from stochastic or induced phenotype shifting of cells within a given population due to higher intrinsic plasticity of PANC-1 cells. It remains to be seen whether these cells exist in a phenotypically stable equilibrium, as observed in breast cancer cell lines [14,16]. In support of this, our PANC-1 clones maintained their E, M or E/M phenotypes upon extended culture and did not re-establish a mixed E/M population similar to parental cultures. In addition, DNA barcoding and high-throughput sequencing may be employed to quantify the extent of intrinsic phenotypic plasticity exhibiting E or M phenotypes, as previously carried out in breast cancer cell clones [52,53].

The close association of TGFβ signaling with the phenotypic plasticity of cancer cells, particularly EMT induction, in skin, breast and lung cancers [8,54], and the variable responses of individual PANC-1 clones to TGFβ1 treatment prompted us to study in more detail the effect on M phenotype shifting in parental PANC-1 cultures. We found that these cultures, despite being QM, or cells with cEMT, can still be induced by treatment with TGFβ1 to attain a more extreme M phenotype (Figure 8A), as evidenced by downregulation of EpCAM, CLDN7, CDH1 and CK19, upregulation of SNAIL, VIM, BGN and SERPINE1, and an increase in invasive activity. This response to TGFβ with respect to EMT and MET has been shown to be principally reversible but to be state-dependent. For instance, in MCF10A cells the transition from E to pEMT was reversible, whereas the transition from pEMT to M was mostly irreversible at high concentrations of TGFβ1 [55] and some cells failed to revert to the E phenotype even when TGFβ was completely withdrawn. It has been speculated that this irreversibility of EMT is most likely due to “extreme” EMT induction [45]. However, using MIA PaCa-2 cells that exhibit such an extreme M phenotype, we were able to partially revert it by treatment with the proinflammatory cytokine cocktail, suggesting that although removal of TGFβ alone might not suffice to reverse EMT, exposure of cells to inducers of MET may be able to accomplish this. Moreover, temporal aspects are crucial, as prolonged TGFβ exposure, mimicking the situation in vivo in the tumor microenvironment, promotes stable EMT in mammary epithelial and carcinoma cells, in contrast to the reversible EMT induced by a shorter exposure. The stabilized EMT was accompanied by stably enhanced stem cell generation and anticancer drug resistance [5]. Mechanistically, a model of cascading switches in phenotypes associated with TGFβ1-induced EMT of MCF10A cells involves two double-negative feedback loops, one between SNAIL and the miR-34 family and another between the related ZEB1 and the miR-200 family. Intriguingly, manipulation of the ZEB/miR-200 balance is able to repeatedly switch cells between E and M states [1], while the induction and maintenance of a stable M phenotype requires the establishment of TGFβ signaling to drive sustained ZEB expression [56].

TGFβ induces broad alterations in splicing patterns by regulating epithelial splicing regulatory proteins (ESRPs) [57], which are also master splice factors during EMT. Of note, knockdown of ESRP1 resulted in the inclusion of exon 3b in alternative splicing of Rac1 mRNA, thereby increasing expression levels of Rac1b [36]. This suggests that the downregulation of RAC1b by TGFβ1 observed here may be mediated through upregulation of ESRP1, a hypothesis that needs to be tested in future studies. Another prominent example of TGFβ-induced differential isoform expression that further enhances the development of TGFβ-dependent EMT programs is represented by fibroblast growth factor receptor (FGFR) [58]. Stable expression of the FGFR-IIb splice variant in M cells promoted the acquisition of E phenotypes, while stable expression of the FGFR-IIc splice variant in E cells elicited M phenotypes reminiscent of those stimulated by TGFβ.

Compared to EMT, the molecular mechanisms and regulators of MET are less well characterized. Besides the IIT mix, which we identified here as a powerful MET inducer (Figure 8A), only few growth factors and signaling pathways are known that promote MET in vivo or in vitro, i.e., BMP7, protein kinase A [59], Notch4 [60] and, interestingly, knockdown of the clock gene, *BMAL1* [31]. GRHL2, a TF that activates *CDH1* and *CLDN4*, and OVOL2 can repress EMT-associated TFs and drive MET. GRHL2 has been shown to regulate epithelial plasticity in pancreatic cancer [61], in sarcomas, where it forms with miR-200 and ZEB1 a gene regulatory network that may push sarcoma cells into an E-like state [62], and in basal-like and claudin-low breast cancer subtypes, in which re-expression of GRHL2 induced N- to ECAD switching consistent with the induction of MET [63]. Of note, GRHL2 also promoted MET programs through its ability to inhibit TGFβ-mediated activation of Smad2/3, which leads to the loss of ZEB1 expression and accompanying upregulation of miR-200 levels [58,63].

The observation of IIT inducing MET was somehow counterintuitive given that TNFα can induce EMT in some systems [64]. Several findings from the literature, however, suggest a scenario in which the MET-inducing effect of proinflammatory cytokines is indirect, involving the intermittent secretion of leukemia-inhibitory factor (LIF) and/or LIF receptor (LIFR). Of note, pancreatic carcinoma cells constitutively express LIF and its heterodimer receptor (LIFR and gp130), and TNFα, IL1β or LIF itself enhanced the expression of LIF mRNA. In agreement with a MET-promoting function, exogenous addition of LIF also promoted cell proliferation in several cell lines, including Hs-700T and PANC-1 [65,66]. Current efforts in our lab are directed at elucidating a possible role of LIF in mediating the MET-inducing function of IIT and further characterizing the E/M states of the various clones using newly identified markers [41].

A significant observation in this study was that treatment of parental PANC-1 cells with TGFβ1 selectively decreased the abundance of RAC1b, while treatment with IIT increased it and additionally downregulated the abundance of RAC1. As mentioned above, high concentrations of TGFβ1 and extended duration of exposure were required to induce stable EMT, and, notably, the same conditions were also required for TGFβ1 to be able to downregulate RAC1b (Figure 7B,C). Together with the data in Figure 5, we propose that the actual RAC1b:RAC1 ratio acts as a switchpoint to determine the particular E or M state and is hence causally involved in determining EMT/MET phenotypes. Previous findings in PANC-1 cells on antagonistic regulation by RAC1b/RAC1 of cell invasion, stemness and cellular plasticity [19,20,40] are clearly in support of this scenario. This would resemble the situation with the two major isoforms of the paired-related homeodomain TF 1 (Prrx1), Prrx1a and Prrx1b, where isoform switching from Prrx1b to Prrx 1a governs EMT plasticity in human PDAC [67]. Specifically, Prrx1b promotes invasion, tumor dedifferentiation and EMT, while Prrx1a drives liver metastatic outgrowth, tumor dedifferentiation and MET [67]. Alternatively, the RAC1b:RAC1 ratio may be exploited as an epiphenomenal biomarker of the EMP state.

Phenotypic plasticity may be exploited for therapeutic interventions particularly to reduce invasion and metastasis because these appear to be more dependent on cellular plasticity than on genetic mutations. Several strategies have been proposed in combating plasticity: (i) driving tumor cells into a “locked” or “irreversible” M state may compromise their ability to colonize distant organs [44,46], (ii) targeting EMP-inducing stimuli which can prevent M conversion, (iii) targeting cells in the M or hybrid E/M state, which can inhibit MET at metastatic sites, (iv) shifting M cells back to the E state [6,11] and (v) converting cancer cells to fat cells [68] or pushing them into apoptosis [69] might be promising approaches in clinical settings. In light of the crucial role of MET in metastatic outgrowth, the potential to effectively target the MET process at sites of metastasis or to block the initial EMT stages that allow the dissemination offers new hope for inhibiting metastatic tumor formation. Our protocol for MET induction possibly involving autocrine induction of LIF also drives attention to targeting inhibitory or stimulatory feedback loops to reduce or promote EMT/MET transitions and to finally curb metastatic potential in vivo (reviewed in Ref. [46]).

In a recent study, Esquer and colleagues [70] identified, isolated and characterized EMT cell populations and their plasticity using a tumor organoid model of EMP in colorectal cancer. These isolated EMT phenotypes displayed different tumorigenic properties and were morphologically and metabolically distinct. Furthermore, stable dual-reporter cell lines generated from colorectal, lung and breast cancers demonstrated a spectrum of EMT cell phenotypes, i.e., E, E/M and M phenotypes. As in our study, these isolated quasi-EMT phenotypes remained stable to spontaneous EMP in the absence of stimuli and during prolonged cell culture but could readily be induced to undergo EMT or MET with growth factors or small molecules. In this context, it will be interesting to see whether the EMT status of a particular clone (E, E/M or M) affects the response to EMT or MET induction by TGFβ1 or IIT, respectively (Figure 8B). Based on the above data on TGFβ1-induced growth inhibition/target gene regulation and on IIT-induced deTD, we believe that they might react very differently.

A 3D high-throughput screening of ~23,000 compounds using dual-reporter QM SW620-derived tumor organoids identified a subset of small molecules that effectively induced MET and decreased malignant properties in isolated QM tumor organoids [70]. These compounds may eventually turn out as suitable drugs to specifically target hybrid/pEMT cells.

While in vitro studies are important to study cellular behavior in the context of phenotypic plasticity and non-genetic ITH, they suffer from the drawback of not presenting the whole landscape of cancer and the genuine EMP spectrum, where cancer cells are infiltrated with stromal cells and reside in an immune microenvironment. Therefore, animal experiments in SCID mouse models need to be performed to evaluate the metastatic potential of the various single cell-derived clones. It also remains to be clarified in more detail how EMP programs cooperate to assist cancer cells through several stages of cancer progression. The results of this study nevertheless expand our understanding of tumor cell plasticity in cancer progression and its contributions towards the development of novel EMP-targeted anti-cancer therapies.

## 4. Material and Methods

### 4.1. Cell Culture and Generation of Single Cell-Derived Clones of PANC-1 and BxPC-3

The PDAC-derived cell lines PANC-1, MIA PaCa-2 and BxPC-3 were maintained in RPMI 1640 basal medium or DMEM supplemented with 10% fetal bovine serum (FBS), 1% Penicillin–Streptomycin–Glutamine (PSG; Thermo Fisher Scientific, Dreieich, Germany) and 1% sodium pyruvate (Merck Millipore, Darmstadt, Germany). In some experiments, cells were stimulated with rec. human (rh) TGFβ1 (#300-023, ReliaTech, Wolfenbüttel, Germany) at 5 ng/mL unless stated otherwise. In some experiments TGFβ1 was used in combination with the ALK5 inhibitor SB431542 (Sigma, Deisenhofen, Germany), or the p38 MAPK inhibitor SB203580 (both from Calbiochem/Merck, Darmstadt, Germany). Six single cell-derived clones of PANC-1 and BxPC-3 were derived by limited dilution.

### 4.2. Quantitative RT-PCR Analysis

Total RNA was isolated from PANC-1 and MIA PaCa-2 cells with an RNAeasy kit (Qiagen, Hilden, Germany). For each sample, 2.5 μg RNA was reverse transcribed with 200 U of M-MLV Reverse Transcriptase and 2.5 μM random hexamers (1 h, 37 °C). Target gene expression was quantified by qPCR on an I-Cycler (BioRad, Munich, Germany) with Maxima SYBR Green Mastermix (Thermo Fisher Scientific) and normalized to the expression of either TATA box-binding protein (TBP) or GAPDH. PCR primer sequences are given in Appendix A.

### 4.3. Immunoblotting

The Western blot procedure was performed as described in detail earlier [38]. In brief, equal amounts of crude proteinaceous extracts from PANC-1, MIA PaCa-2 or BxPC-3 cells (5–7 preparations harvested at different times during continuous culture) were fractionated by SDS-PAGE on mini-PROTEAN TGX any-kD precast gels, or TGX Stain-Free FastCast gels (BioRad) and blotted to PVDF membranes. Membranes were blocked with nonfat dry milk or bovine serum albumin and incubated with primary antibodies to CLDN4 (clone 3E2C1, #18-7341, Zymed Laboratories, South San Francisco, CA, USA), ECAD (#610181, BD Transduction Laboratories, Heidelberg, Germany), RAC1 (#610650, BD Transduction Laboratories), RAC1b (#09-271, Merck Millipore), SNAIL (#3895, Cell Signalling Technology, Frankfurt/Main, Germany) or VIM (clone V9, #V6630, Sigma-Aldrich, Steinheim, Germany), and either GAPDH (Cell Signalling Technology) or HSP90 (#13119, Santa Cruz, Heidelberg, Germany) as loading controls. Chemoluminescent detection of proteins was performed with a ChemiDoc XRS+ System and Image Lab Software (BioRad) using Amersham ECL Prime Detection Reagent (GE Healthcare, Munich, Germany). The ChemiDoc XRS+ system also permitted the densitometric quantification of signal intensities from underexposed autoradiographs. Densitometric readings for the proteins of interest were normalized to the total amount of protein in the same lane (when using the TGX Stain-Free FastCast gels), or to bands for GAPDH or HSP90 (when using the mini-PROTEAN TGX any-kD precast gels). The original Western blot images can be found in Appendix A.

### 4.4. Cell Proliferation and Viability Assays

PANC-1 cells were seeded at 10,000 cells per 12-well (for the cell counting assay) or at 1000 cells per 96-well (for the resazurin assay) plate and treated on the next day with rhTGFβ1 (5 ng/mL) or vehicle, for 72 h or 6 d, respectively. Following trypsinization, cells from three wells were counted either manually using a Neubauer chamber or with the Cedex XS device (Roche Diagnostics, Mannheim, Germany). The resazurin (Alamar blue) assay was performed in a 96-well format as suggested by the supplier (Thermo Fisher Scientific). Each condition was tested in six technical replicates. Two-thousand cells per well were seeded in a 96-well plate, 12 wells for each clone, 6 control and 6 treated with TGFβ1 for 6d. Resazurin was diluted with medium (DMEM-high glucose + GlutaMAX) and 20 µL Resazurin was added to each well. The plate was wrapped in aluminium foil and incubated for 2 h in an incubator. Fluorescence was measured at 530 (excitation) and 600 (emission). Mean intensities with blank (medium only) subtracted.

### 4.5. Colony Formation Assay

The procedure for the colony formation assay was described in detail elsewhere [20]. Briefly, 400 cells were seeded per 6-well plate in standard growth medium in a triplicate setup. After 10–12 d, the medium was removed, cells were rinsed once in PBS and fixed in 4% paraformaldehyde for 1 h. Following staining with crystal violet, cells were air-dried and colonies consisting of more than 50 cells were counted manually under a microscope.

### 4.6. Measurement of Circadian Rhythmicity

PANC-1 cells were kept in RPMI 1640 basal medium without phenol red and supplemented as specified above. For luminescence experiments, 200,000 cells were reverse transfected using Lipofectamine 3000 (Thermo Fisher Scientific) according to the manufacturer’s instructions. In brief, 3 µg of *Bmal1::Luc* plasmid (pABpuro-BluF, Addgene #46824) was added together with 6 µL of P3000 in a tube containing 150 µL of Opti-MEM (Thermo Fisher Scientific). In another tube, 150 µL of Opti-MEM was combined with 3 µL of Lipofectamine 3000. Both tubes were mixed and left at RT for 10 min. Then, 310 µL solution was added to each dish, which was followed by the addition of cells resuspended in medium. Twenty-four hours later, cells were synchronized with dexa (200 nM, Sigma-Aldrich), forskolin (10 µM, Sigma-Aldrich) or 50% horse serum (Thermo Fisher Scientific). As a positive control, GFP plasmid was used to validate the efficiency of the transfection (data not shown). After 2 h, cells were washed with PBS and media containing 200 µM of luciferin (AppliChem, Darmstadt, Germany) was added. Dishes were sealed with parafilm and placed into a Lumicycle luminometer at 32.5 °C (Actimetrics, Wilmette, IL, USA). Every 10 min, baseline-subtracted (24 h running average) bioluminescence data were collected and exported to Prism 9 (GraphPad, San Diego, CA, USA). Rhythmicity was assessed using the Circasingle algorithm [71], using the average of each reading per hour and by considering amplitude as decay factor for clones P3D2, P3D10 and MIA PaCa-2. Period values were not predefined. Rhythmicity was confirmed when a *p*-value < 0.05 was achieved. For rhythmic data, a dampened sine curve was fitted using a robust fit with a period >18–20 h.

### 4.7. In Vitro Transdifferentiation to Endocrine Progenitors and Insulin-Expressing Cells and Induction of MET

Ductal-to-endocrine transdifferentiation of PANC-1 cells to cells with elevated *NEUROG3* and *INS* expression and the induction of MET was achieved by employing a previously published protocol [30]. The procedure employed a combination of the proinflammatory cytokines INFγ (100 ng/mL), IL1β (25 ng/mL) and TNFα (50 ng/mL) designated “IIT” in standard growth medium with some modifications that were outlined in detail elsewhere [19,20].

### 4.8. Migration and Invasion Assays

The invasive activity of PANC-1 cells was determined with xCELLigence^®^ technology (ACEA Biosciences, San Diego, supplied by OLS, Bremen, Germany) as outlined in detail in previous publications [39,40]. The lower side of the CIM plate-16 porous membrane was coated with a 1:1 mixture (*v*/*v*) of collagen I and collagen IV (30 μL) to facilitate adherence of the cells and thus enhance the duration of signal recording. Prior to cell seeding, the surface of the upper chamber was covered with a thin monolayer of 5% (*v*/*v*) growth factor-reduced Matrigel (BD Biosciences, Heidelberg, Germany) diluted 1:20 with basal medium, as detailed elsewhere [72]. After the Matrigel solification, each well was loaded with 60,000 or 80,000 cells in standard growth medium (see above). Cells were allowed to settle in the laminar flow hood for 30 min at RT, after which the assay was started and run for 24–48 h. Data acquisition (with signal recording every 15 min) and analysis was performed with the RTCA software (version 1.2, ACEA).

### 4.9. Statistical Analysis

Statistical significance was calculated using either the two-tailed, unpaired Student’s *t*-test or the Wilcoxon test to assess interassay variability, and data were given as means ± SD or SEM. Results were considered significant at *p* < 0.05 (denoted by one asterisk). In some experiments, higher levels of significance were calculated and denoted by two (*p* < 0.01) or three (*p* < 0.001) asterisks.

## 5. Conclusions

The establishment of ITH and cellular plasticity is driven by EMT and MET programs that jointly control the dynamic interconversions of E cells to M cells along a spectrum of intermediate states. Targeting these states might be a promising approach in clinical settings to combat invasion, metastasis and drug resistance. Several specific strategies have already been proposed; however, before these can be applied to patients a better understanding of the mechanisms responsible for MET induction, maintenance of the E/M phenotype(s) and the contribution that intermediate states of the EMT spectrum make to tumor evolution is required. Depending on the clinical scenario, MET-inducing/stabilizing factors may inhibit metastasis if they block the initial EMT stages that allow tumor cells to disseminate from the primary tumor or curb metastatic outgrowth at distant sites. Our protocol for MET induction in QM-type PDAC cells may be adopted by other researchers to screen for these factors in an effort to develop a new class of EMP-directed biologicals for pancreatic cancer treatment.

## Figures and Tables

**Figure 1 cancers-14-02057-f001:**
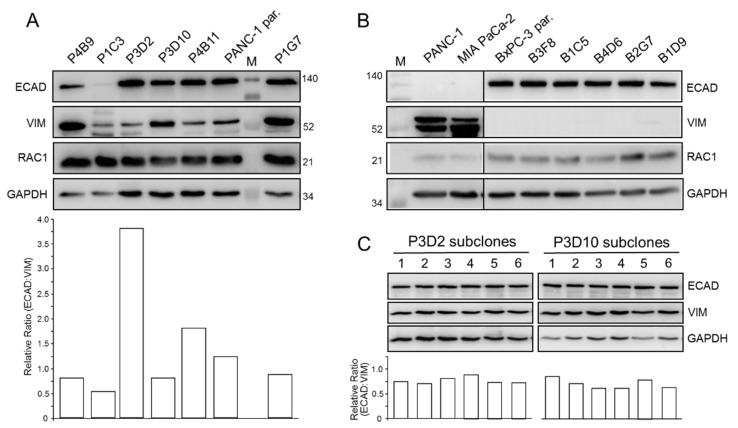
Heterogeneity and phenotypic stability of single cell-derived clones of PANC-1. (**A**) Immunoblot analysis of the EMT-related biomarkers ECAD, VIM and RAC1 in single cell-derived clones of PANC-1. The housekeeping protein GAPDH served as a loading control. The blots shown are representative of 5–7 samples per clone taken at different time points during continuous culture. The graph below the blot depicts the relative ratios of ECAD:VIM band intensities derived from densitometric readings. The numbers on the right-hand side denote molecular weights in kDa. (**B**) As in (**A**), except that single cell-derived clones of BxPC-3 cells were analyzed. (**C**) ECAD and VIM expression in subclones of PANC1-clones P3D2 and P3D10. Single cell-derived subclones of P3D2 and P3D10 were generated by limited dilution and monitored for ECAD and VIM expression, as described in (**A**). The relative ECAD:VIM ratios of band intensities from the various subclones are presented underneath the blots.

**Figure 2 cancers-14-02057-f002:**
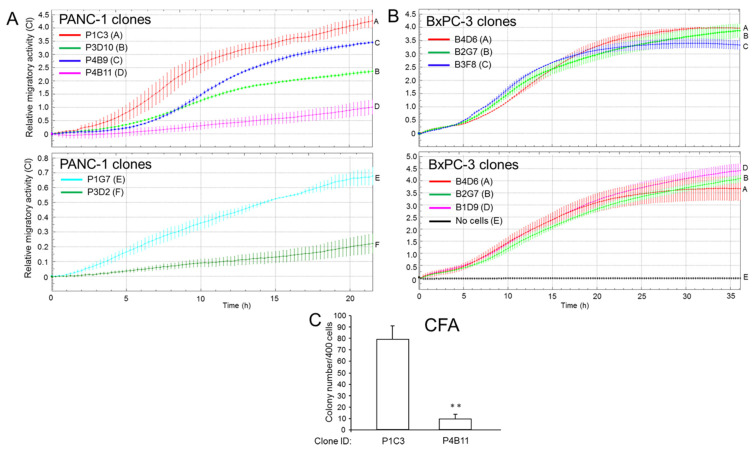
Heterogeneity of individual PANC-1 clones in invasive activities and stemness features. (**A**) Real-time cell migration assays of six individual clones of PANC-1. (**B**) As in (**A**), except that the migration assays were performed with four individual clones of BxPC-3. Data in (**A**,**B**) are the mean ± SD of quadruplicate wells and are representative of three assays. (**C**) CFA of the indicated PANC-1 clones. Four-hundred cells of each clone were seeded per six-well plates and incubated for 10–12 d, after which the colonies formed were fixed and stained and the colonies consisting of >50 cells were counted manually under a microscope. Data are the means ± SD of triplicate wells. The asterisks indicate significance. ** *p* < 0.01 (two-tailed unpaired Student’s *t*-test).

**Figure 3 cancers-14-02057-f003:**
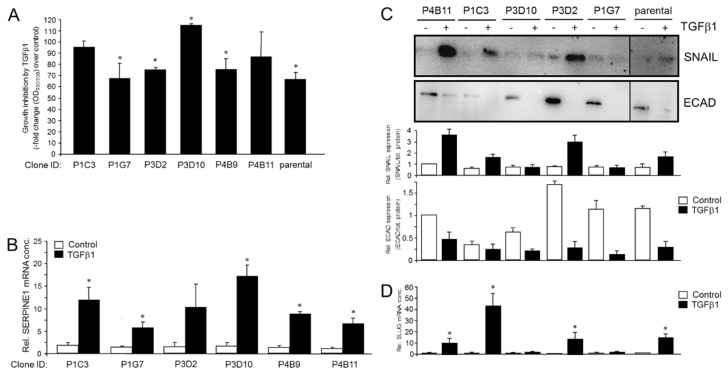
Heterogeneity in cellular responses of individual PANC-1 clones to stimulation with TGFβ1. (**A**) The indicated PANC-1 clones were treated for six days with TGFβ1 in standard growth medium prior to the resazurin assay. Data are the means ± SD of three independent assays. The number of dead cells in all assays was <1%. (**B**) Response of *SERPINE1* to stimulation with TGFβ1. Cells analyzed in (**A**) were subjected to RNA isolation and qPCR for *SERPINE1*. Data shown are the mean ± SD of three assays. (**C**) Immunoblot analysis of SNAIL, and ECAD as control, of the indicated PANC-1 clones after a 72 h treatment with TGFβ1. Quantitative data depicted as graphs underneath the blots (mean ± SD of three technical replicates) were derived from densitometric readings of signal intensities normalized to total protein loading and are representative of three experiments. (**D**) Basal and TGFβ1-induced expression of SLUG. The five PANC-1 clones and parental cells from (**C**) were subjected to qPCR for *SNAIL2*. Data are representative of three assays and are the mean ± SD of triplicate wells. Asterisks (*) denote a significant difference.

**Figure 4 cancers-14-02057-f004:**
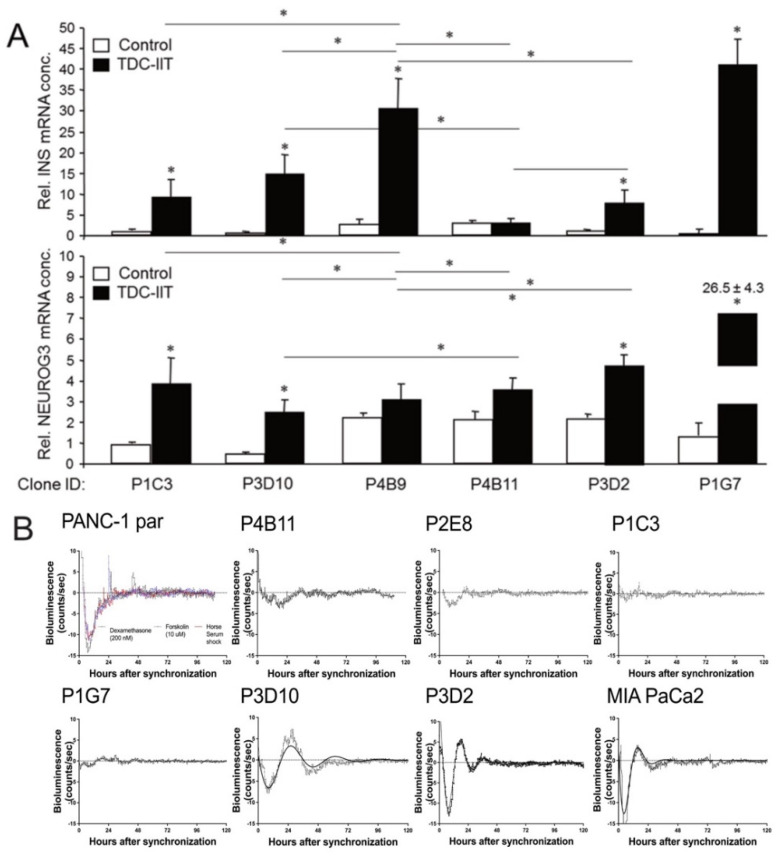
PANC-1 clones and heterogeneity with respect to deTD and circadian rhythmicity. (**A**) The indicated PANC-1 clones were subjected to deTD culture for 5 days in normal growth medium supplemented with 100 ng/mL IFNγ, 25 ng/mL IL1β and 50 ng/mL TNFα (TDC-IIT). Following lysis and RNA extraction, *INS* and *NEUROG3* expression was determined by qPCR and C_t_ values normalized with those for GAPDH. The assay shown (means ± SD of three replicates) is representative of three experiments. Significant differences (*p* < 0.05) are marked by asterisks (*). (**B**) *Bmal1::Luc* bioluminescence reporter assays in parental (par) PANC-1 cells and subclones. Each clone is presented separately. For PANC-1, each applied synchronization treatment is depicted in a different color, as indicated (dexa: black; forskolin: blue; horse serum shock: red). The 24 h baseline subtracted bioluminescence data are shown as mean ± SEM. Data are pooled from two independent experiments with 4–8 replicates each. Dampened sine curves were fitted for rhythmic clones for rhythm parameter determination (Appendix A).

**Figure 5 cancers-14-02057-f005:**
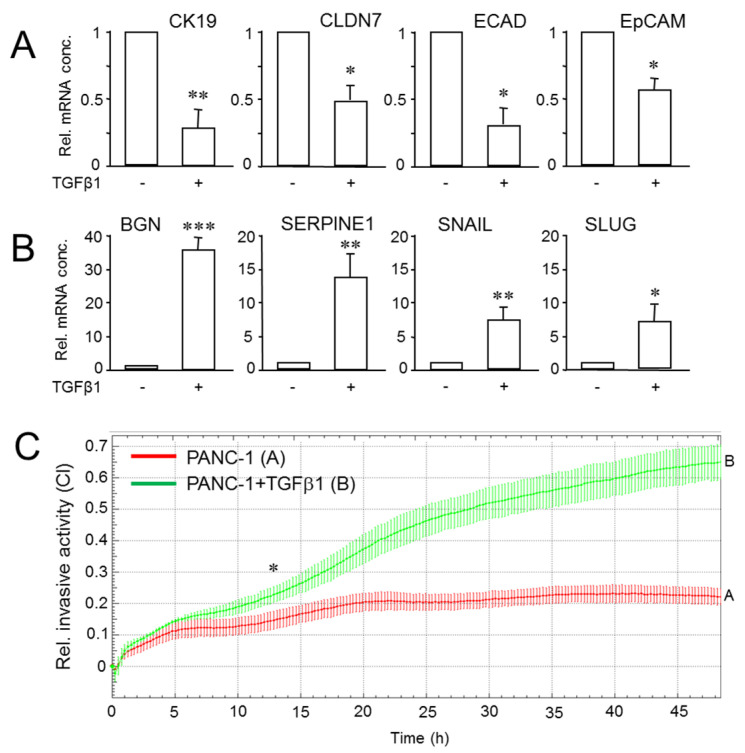
PANC-1 cells can be shifted towards a more mesenchymal and invasive phenotype by activation of a TGFβ transcriptional program. (**A**) Parental PANC-1 cultures were treated with TGFβ1 for 24 h followed by qPCR analysis of CK19, CLDN7, ECAD and EpCAM. The data shown (means ± SD of three replicates) are representative of three experiments. Significant differences (*p* < 0.05) are marked by asterisks (*). (**B**) As in (**A**), except that primers for BGN, PAI-1, SNAIL1 and SLUG were used. C_t_ values were normalized with those for GAPDH in the same samples. (**C**) PANC-1 cells were subjected to real-time cell invasion assay on an xCELLigence platform in the presence or absence of TGFβ1 (5 ng/mL). Data are representative of three assays and are the means ± SD of triplicate wells. The asterisk marks the earliest time point with significant differences between TGFβ1-treated and untreated control cells. ** *p* < 0.01, *** *p* < 0.001.

**Figure 6 cancers-14-02057-f006:**
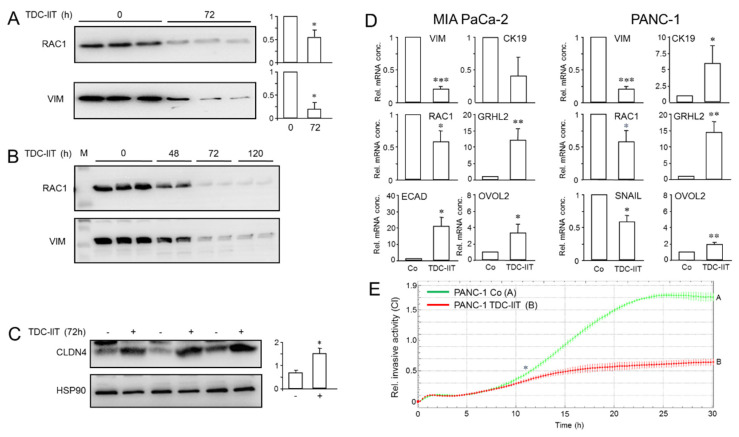
PANC-1 and MIA PaCa-2 cells can be shifted towards a more epithelial phenotype. (**A**) PANC-1 cells were subjected to TD culture with IIT (TDC-IIT) for 72 h followed by immunoblot analysis of the indicated proteins. The assay shown (means ± SD of three replicates) is representative of three experiments. The graphs to the right show data quantification from three assays (*n* = 3) based on densitometric readings of signal intensities normalized to the total protein content of the corresponding lane (Appendix A). Significant differences (*p* < 0.05) are marked by asterisks (*). (**B**) As in (**A**), except that MIA PaCa-2 cells were used. Again, signal intensities of RAC1 and VIM were normalized to values for total protein loading in the same lane (Appendix A). (**C**) As in (**A**), except that CLDN4 was detected and signal quantification shown in the graph on the right-hand side. (**D**) QPCR analyses of the indicated genes in MIA PaCa-2 and PANC-1 cells. Values were normalized with the housekeeping genes GAPDH or TBP in the same samples. Data are the means ± SD of three experiments (*n* = 3). Significant differences (*p* < 0.05) are marked by asterisks (*). The assays shown are representative of three experiments. Significant differences are marked by asterisks. *, *p* < 0.05, **, *p* < 0.01, *** *p* < 0.001. (**E**) PANC-1 cells were pretreated with IIT for 48 h and thereafter subjected to real-time cell invasion assay (in the absence of IIT) using Matrigel as barrier. Data are the means ± SD of three parallel wells and are representative of three assays. The asterisk (*) marks the time point at which data first become significant.

**Figure 7 cancers-14-02057-f007:**
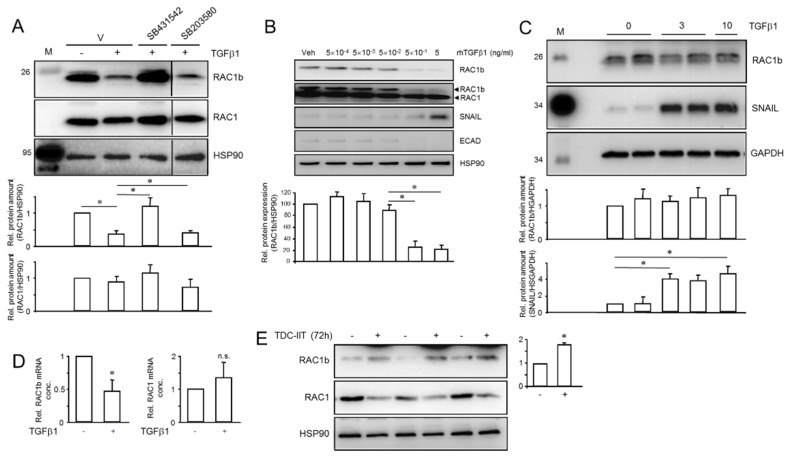
Treatment with TGFβ1 downregulates endogenous but not ectopically expressed RAC1b in PANC-1 cells. (**A**) PANC-1 cells were treated or not for 3 d with 5 ng/mL of TGFβ1 in the absence or presence of vehicle (DMSO) or the indicated small molecule inhibitors, followed by lysis and sequential immunoblotting of RAC1b, RAC1 and HSP90 as a loading control. Data quantification by densitometric analysis of band intensities plotted relative to untreated control cells represents the mean ± SD of three experiments. Asterisks (*) indicate a significant difference. The horizontal lines between lanes 4 and 5 of the blots denote removal of irrelevant lanes. (**B**) PANC-1 cells were treated, or not, for 120 h with the indicated concentrations of rec. human (rh) TGFβ1 followed by sequential immunoblotting of RAC1b, RAC1, SNAIL and ECAD. The graph below the blot provides quantitative data for RAC1b derived from densitometric readings of RAC1b and HSP90 (mean ± SD, *n* = 3). (**C**) PANC1-1 cells ectopically expressing HA-tagged RAC1b were left untreated or were treated for 3 d or 10 d with TGFβ1 followed by immunoblot analysis of RAC1b, SNAIL and GAPDH as loading control. Please note unaltered expression of ectopic RAC1b and induction of SNAIL as a control for TGFβ1 biological activity. Data are the means ± SD of three replicates and are representative of three experiments performed in total. (**D**) Effect of a 24 h treatment with TGFβ1 on RAC1b (left-hand graph) or RAC1 (right-hand graph) steady-state mRNA levels as measured by qPCR (mean ± SD of three experiments). The asterisk indicates a significant difference; n.s., not significant. (**E**) PANC-1 cells were subjected to TDC-IIT for 72 h followed by immunoblot analysis of RAC1 and RAC1b. The assay shown (means ± SD of three replicates each) is representative of three experiments. The graph to the right shows RAC1b signal quantification based on densitometric readings from three assays (*n* = 3). The asterisk (*) denotes a significant difference (*p* < 0.05).

**Figure 8 cancers-14-02057-f008:**
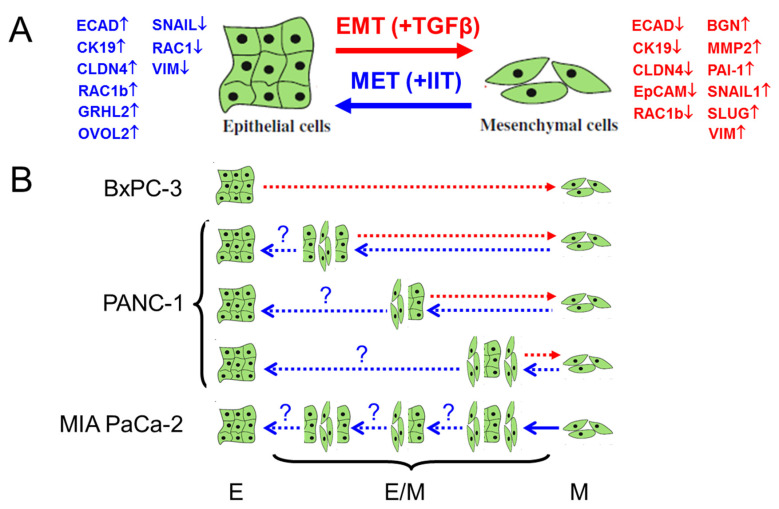
Cartoon illustrating heterogeneity in PANC-1 cells with respect to EMT/MET phenotypes. (**A**) Principal induction of EMT (red lines), e.g., by growth factors such as TGFβ, or induction of MET (blue lines), e.g., by proinflammatory cytokines INFγ, IL1β and TNFα (IIT), and the associated changes in marker expression (↑ and ↓ denote up- and downregulation, respectively). (**B**) Schematic representation of the different EMT phenotypes present in parental cultures of PANC-1 based on marker expression and functional properties of the six clones. The stippled lines denote the principal ability of the hybrid phenotypes for further dedifferentiation towards a more complete/extreme M phenotype, or redifferentiation to the previous E state. With regard to the latter event, it is currently unclear whether IIT treatment can reprogram MIA PaCa-2 cells to an extreme E phenotype or merely to one of several hybrid E/M states (denoted by question marks). It also remains to be investigated whether all of the hybrid E/M phenotypes present in PANC1 cultures respond to TGFβ1 or IIT with induction of these differentiation events and whether their concentration or duration of exposure is critical. BxPC-3 cells are unlikely to harbor E/M or M-type cells as they are VIM negative, while MIA PaCa-2 cells are unlikely to contain E or E/M-type cells as they are ECAD negative.

## Data Availability

The data presented in this study are available in this article (and Appendix A).

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
