# Peer review of "The Quasimesenchymal Pancreatic Ductal Epithelial Cell Line PANC-1—A Useful Model to Study Clonal Heterogeneity and EMT Subtype Shifting"

_cancers, 2022, doi:10.3390/cancers14092057_

Round 1

Reviewer 1 Report

Ungefroren et al isolated single clones of the pancreatic cancer cell line PANC-1 and showed that they possess different degrees of epithelial- and mesenchymal- like characteristics. The individual clones also responded differently to TGFb treatment. Their data suggest that the PANC-1 cell line and the isolated individual clones could represent a useful model to study epithelial to mesenchymal plasticity.

  1. RAC1 expression in PANC1 E or M clones is shown in figure 1A. How is the expression of RAC1B in these clones?
  2. A correlation plot between the migratory activity (fig. 2A) and ECAD/VIM ratio (fig. 1A) across the clones would be nice to show the association between E/M status and the migratory activity.
  3. Why were not all clones analyzed in the colony forming assay (fig 2B)?
  4. Does the sensitivity to TGFB1 (figure3) correlate with the E/M status of clones?
  5. Does the sensitivity to deTD (figure4A) correlate with the E/M status of clones?
  6. Do the single cell clone remain in the corresponding E, E/M, or M state if cultured for longer period? Is the phenotype stable or do they re-establish a mixed E/M population similar to parental? To analyze this, a flow cytometry experiment could be performed to assess the frequency of ECAD and VIM positive cells in individual clones and eventual change in the frequencies over time.

7. Loading control WBs are missing in fig 3C, 4A, and 4B

Author Response

Dear Editor, dear Wendy:

This letter of submission is accompanied by our revised manuscript entitled:

“The quasimesenchymal pancreatic ductal epithelial cell line PANC-1 – a useful model to study clonal heterogeneity and EMT subtype shifting”

We are indebted to the reviewers for their enthusiastic and valuable comments and suggestions and have incorporated almost all of these into the revised version of our manuscript (highlighted in the “track changes” mode). We believe that the reviewers’ critiques have substantially improved the quality of our manuscript and we are looking forward to its final acceptance in Cancers.

Sincerely yours,

Hendrik Ungefroren

Reviewer 1

Comment #1

Ungefroren et al isolated single clones of the pancreatic cancer cell line PANC-1 and showed that they possess different degrees of epithelial- and mesenchymal- like characteristics. The individual clones also responded differently to TGFb treatment. Their data suggest that the PANC-1 cell line and the isolated individual clones could represent a useful model to study epithelial to mesenchymal plasticity.

1. RAC1 expression in PANC1 E or M clones is shown in figure 1A. How is the expression of RAC1B in these clones?

Response: This is a good question given that we study RAC1b expression later in this manuscript. We have detected RAC1b by immunoblotting, quantified its expression along with that of RAC1 and have calculated the ratio of RAC1b/RAC1 protein levels (now mentioned in the second paragraph of section 2.1.). We found the protein expression of RAC1b among the clones to be more variable than that of RAC1. When correlated with migratory activity, we observed that clones with low migratory activity (P3D2, P4B11, P1G7) exhibited a high RAC1b:RAC1 ratio, while in the three clones with high migration potential (P1C3, P4B9, P3D10) this ratio was low(er) and thus behaved like the ECAD:VIM ratio. This is in accordance with our earlier studies in breast cancer cell lines (PMID: 33567745). These data are now shown in the new Figures S2 (quantification) and Figure S4B (correlation of relative RAC1b:RAC1 ratios with migratory activities).

2. A correlation plot between the migratory activity (fig. 2A) and ECAD/VIM ratio (fig. 1A) across the clones would be nice to show the association between E/M status and the migratory activity.

Response: We perfectly agree with the reviewer and, as suggested, have correlated both parameters in the new Figure S4A.

3. Why were not all clones analyzed in the colony forming assay (fig 2B)?

Response: We decided to perform this assay only with two PANC-1 clones, one representing the group of strong migrators (P1C3) and another one from the group of weak migrators (P4B11).

4. Does the sensitivity to TGFB1 (figure3) correlate with the E/M status of clones?

Response: This is an interesting issue. When assessed on the SERPINE-1 response there seems indeed to be a correlation since the most M-type clones (= low ECAD/VIM ratio) exhibited a greater response than the E-type clones P1G7 and P4B11, as well as the third E-type clone (P3D2), for which the TGFb1 effect was not even statistically significant. We have added a sentence to the second paragraph of section 2.3. to indicate this.

5. Does the sensitivity to deTD (figure4A) correlate with the E/M status of clones?

Response: The situation is similar to the TGFb1 sensitivity with the notable exception of clone P1G7, which exhibits by far the greatest response to deTD although it is considered an E-type clone. We do not currently have a molecular explanation for this.

6. Do the single cell clone remain in the corresponding E, E/M, or M state if cultured for longer period? Is the phenotype stable or do they re-establish a mixed E/M population similar to parental? To analyze this, a flow cytometry experiment could be performed to assess the frequency of ECAD and VIM positive cells in individual clones and eventual change in the frequencies over time.

Response: This is a very good question and we thank the reviewer for motivating us to analyse this in more detail. In order to make results comparable with those in Figure 1, we have used immunoblotting rather than the suggested flow cytometry. Two strategies were followed to tackle this issue:

i) a comparison of early and late passage cells from 3 clones for expression of ECAD and VIM did not reveal significant changes in expression of either protein (data not shown), and ii) by limited dilution, we generated single cell-derived subclones of three clones (P3D2, P3D10 and P4B11) and compared 5-6 of these with respect to growth pattern/cell morphology and expression of ECAD and VIM by immunoblotting. Again, both the morphology and ECAD and VIM expression was quite homogenous among the subclones. Together, this indicated that the phenotype of the various clones is stable and they apparently do not re-establish a mixed E/M population similar to parental cultures. These data have been added to Figure 1 as new panel C. The former panel C (SYP qPCR in PANC-1 clones) has been moved to the Suppl. material as new Figure S3.

7. Loading control WBs are missing in fig 3C, 4A, and 4B.

Response: We observed in some experiments that the 3 or 5-day IIT treatment of PANC-1 or MIA PaCa-2 cells (the reviewer likely refers here to Figs. 6A and B rather than 4A and B) also resulted in less intense bands of the housekeeping proteins GAPDH and HSP90, although we took care to load the same amount of total protein on each lane in all experiments. It should be emphasized that this was not a general inhibitory effect on all proteins as bands for claudin-4 in Fig. 6C become more intense! We therefore decided to normalize the signals for RAC1 and VIM in Figs. 6A and B (and those for SNAIL and ECAD in Fig. 3C) to the total protein content loaded on the respective lane. These blots are now supplied in Figure S12. From the distribution of bands it is apparent that IIT induces some proteins to become more abundant and others to become less abundant.

Additional changes made

  1. Due to the addition of various new Supplementary figures, the numbering of the original Suppl. figures has been updated as follows:

Figure S1A: Morphology and growth patterns of single cell-derived clones of PANC-1.

Figure S1B (new): Morphology and growth patterns of single cell-derived subclones of PANC-1 clones P3D2 and P4B11.

Figure S2 (new): Quantification of RAC1 and RAC1b protein levels in PANC-1 single clones.

Figure S3 (formerly Fig. 1C): Expression analysis of the neuroendocrine marker SYP in the indicated PANC-1 clones.

Figure S4 (new): Correlation of the ECAD/VIM and RAC1b/RAC1 ratios with migratory activities in six individual PANC-1 clones.

Figure S5 (new): Migratory activity of parental PANC-1 or BxPC-3 cells compared to two single cell-derived clones.

Figure S6 (in part new, formerly S2): Proliferative responses of individual PANC-1 clones under basal conditions and after stimulation with TGFβ1 as determined by cell counting assay.

Figure S7 (formerly S3): Rhythmic parameters evaluation by CircaSingle.

Figure S8 (formerly S4): Morphology and growth pattern of parental PANC-1 cultures treated with TGFβ1.

Figure S9 (formerly S5): Immunoblot analysis of SNAIL, RAC1 and VIM in parental PANC-1 cells after treatment with IIT.

S10-S14: Uncropped blots.

  1. For the sake of consistency, the terms “epithelial” and “mesenchymal” have been replaced by their abbreviations “E” and “M”, respectively.
  2. Two new references have been added to the References section.
  3. In Figure 1A and 1C, the labeling of the ordinates has been changed from “Ratio ECAD/VIM” to “Relative ratio ECAD:VIM” since the ECAD and VIM signals were detected independently with different exposure times.

Reviewer 2 Report

The authors provide a useful model to study EMP after performing experiments on pancreatic ductal adenocarcinoma cells. They have generated single cell-derived clonal cultures from PANC-1 cells showig a different classification of  EMT phenotypes with various colony forming/invasive activities, TGF-b response and circadian clock function. Thus, the results are able to expand the understanding on tumor cell plasticity in tumor progression for future EMP-targeted anti-cancer therapies. Indeed, nowadays there is a lack of cellular models to analyse EMP. In this way the findings for MET induction in PANC cells may be adopted by other researchers to screen for these factors in an effort to develop a new class of EMP-directed biologicals for pancreatic cancer treatment, above all at the light of multidrug resistance scenario.

The data of the study are well presented and described in details. Figures are exhaustive and references are appropraite. The manuscript is well written, but the auhors should avoid the repetitions in text, in  simple summary, in abstract and conclusion.

Author Response

Dear Editor, dear Wendy:

This letter of submission is accompanied by our revised manuscript entitled:

“The quasimesenchymal pancreatic ductal epithelial cell line PANC-1 – a useful model to study clonal heterogeneity and EMT subtype shifting”

We are indebted to the reviewers for their enthusiastic and valuable comments and suggestions and have incorporated almost all of these into the revised version of our manuscript (highlighted in the “track changes” mode). We believe that the reviewers’ critiques have substantially improved the quality of our manuscript and we are looking forward to its final acceptance in Cancers.

Sincerely yours,

Hendrik Ungefroren

The authors provide a useful model to study EMP after performing experiments on pancreatic ductal adenocarcinoma cells. They have generated single cell-derived clonal cultures from PANC-1 cells showing a different classification of EMT phenotypes with various colony forming/invasive activities, TGF-b response and circadian clock function. Thus, the results are able to expand the understanding on tumor cell plasticity in tumor progression for future EMP-targeted anti-cancer therapies. Indeed, nowadays there is a lack of cellular models to analyse EMP. In this way the findings for MET induction in PANC cells may be adopted by other researchers to screen for these factors in an effort to develop a new class of EMP-directed biologicals for pancreatic cancer treatment, above all at the light of multidrug resistance scenario.

The data of the study are well presented and described in details. Figures are exhaustive and references are appropraite. The manuscript is well written, but the auhors should avoid the repetitions in text, in simple summary, in abstract and conclusion.

Response: We thank this reviewer for his enthusiastic comments. We have carefully checked the mentioned parts of our manuscript for the presence of repetitions and have removed a few. In other cases, however, we believe that repeating some crucial points in introductory sentences will enhance understanding of the text and we would therefore prefer to leave the text as its stands.

Additional changes made

  1. Due to the addition of various new Supplementary figures, the numbering of the original Suppl. figures has been updated as follows:

Figure S1A: Morphology and growth patterns of single cell-derived clones of PANC-1.

Figure S1B (new): Morphology and growth patterns of single cell-derived subclones of PANC-1 clones P3D2 and P4B11.

Figure S2 (new): Quantification of RAC1 and RAC1b protein levels in PANC-1 single clones.

Figure S3 (formerly Fig. 1C): Expression analysis of the neuroendocrine marker SYP in the indicated PANC-1 clones.

Figure S4 (new): Correlation of the ECAD/VIM and RAC1b/RAC1 ratios with migratory activities in six individual PANC-1 clones.

Figure S5 (new): Migratory activity of parental PANC-1 or BxPC-3 cells compared to two single cell-derived clones.

Figure S6 (in part new, formerly S2): Proliferative responses of individual PANC-1 clones under basal conditions and after stimulation with TGFβ1 as determined by cell counting assay.

Figure S7 (formerly S3): Rhythmic parameters evaluation by CircaSingle.

Figure S8 (formerly S4): Morphology and growth pattern of parental PANC-1 cultures treated with TGFβ1.

Figure S9 (formerly S5): Immunoblot analysis of SNAIL, RAC1 and VIM in parental PANC-1 cells after treatment with IIT.

S10-S14: Uncropped blots.

  1. For the sake of consistency, the terms “epithelial” and “mesenchymal” have been replaced by their abbreviations “E” and “M”, respectively.
  2. Two new references have been added to the References section.
  3. In Figure 1A and 1C, the labeling of the ordinates has been changed from “Ratio ECAD/VIM” to “Relative ratio ECAD:VIM” since the ECAD and VIM signals were detected independently with different exposure times.

Reviewer 3 Report

In the manuscript by Ungefroren and colleagues, the authors present the well-known and widely used pancreatic cancer cell line PANC1 as tool to analyze and experimentally modulate partial EMT states. Such tools are very valuable to address a very important research question about the function of epithelial-mesenchymal plasticity on tumor progression and metastasis. They used WB, qRT-PCR and invasion assays to characterize PANC1 heterogeneity. In addition, the response of derived single-cell clones and the parental cell line to TGFbeta and a cytokine cocktail of transdifferentiation (IFNγ, IL1β and TNFα) was analyzed. They found that single cell clones showed differential gene expression of some EMT markers and varying invasion capacities and TGFbeta response. Moreover, the parental line was used to further dissect changes in gene expression upon TGFbeta and IFNγ, IL1β and TNFα treatment, resulting in shifts towards more mesenchymal and epithelial phenotypes, respectively. A model that shows a partial EMT state under steady-state conditions and can be experimentally shifted towards both mesenchymal and epithelial states is of high value. However, despite this importance, I miss a bit the novelty aspect of the findings to merit publication in “cancers”. For example, heterogeneity of Panc1 and other cell lines with respect to EMP was shown before, e.g. in (21). The presented data are rather preliminary and the manuscript lacks a thread. To me it remains elusive whether the authors want to analyze the heterogeneity feature of PANC1 cells (in contrast to BxPC3) or to introduce a tool for EMP research. For the former part, the authors fail to correlate the clonal heterogeneity with specific features along the EMP/EMT axis. Most experiments either need some more in depth analysis or describe observations that are not new. More specific points are listed below:

  1. In general, the choice of EMT/EMP markers are chosen a bit random (like SYP, RAC1, Snail, Serpine, Claudin4 (but not other EMT-TFs). Why is SYP only analyzed on transcript level? More markers need to be assessed to characterize the clone identity in respect to the EMP continuum (e.g. Snail/E-cad expression levels in steady-state, TGFbeta response, IIT response, migration/invasion: are these features correlated?)
  2. The order of the clones and in general the order of WB (e.g. Fig. 6: Rac1 top, Vim bottom (A) and vice versa (B)) is changing multiple times, which makes it hard to compare
  3. Figure S1: Phase contrast image of the parental cell line should be shown as well
  4. 2: parental PANC1 and BxPC3 should be included
  5. 3A etc.: is fold-change or percentage shown here?
  6. Figure 3: The authors aim on analyzing proliferation. However, without assessing cell death, the data only show relative cell numbers over time, but no indication about proliferative activities. What about the differences in cell numbers/proliferation under steady-state conditions? This aspect is very important otherwise relative changes do not mean much. Moreover, since BxPC3 clones did not show EMP differences, it would be interesting to see their performance here as well.
  7. Figure 4B: I am puzzled by the rational of this experiment. The results are completely different to the original observation by (30), maybe because of highly variable plasmid DNA uptake per cell by transient transfection in comparison to transduction. In any case, I do not see why these experiments are crucial for identifying clonal variation and describing EMP differences. Shouldn't these graphs also include a measurement without any synchronization (ctrl)? What is the conclusion of this experiment with regard to EMP?
  8. Figs 5-7: After characterization of the single cell clones, why do the authors switch back to the parental cell line?
  9. Figure 6: The IIT treatment should also be done for the single cell clones! To me, the important question is, are Panc1 cells representing a heterogeneity that is reflected in the clonal expansion, do the clones all respond in a similar/different way to the various treatments, like the parental cells, but with varying extend? Do these clones represent specific states within the EMP continuum?
  10. Figure 6D: The data of CK19 and E-cad should be shown even if there is no difference.
  11. Figure 7B: what do the authors conclude from the ALK5/MAPK inhibitor treatment? Is non-canonical TGFbeta signaling responsible for alternative RAC1 splicing? The authors state that “relatively high” TGFbeta doses are necessary for a robust RAC1b decrease. However, 5 ng/ml were used throughout the paper and is also used by many others. So, the term "relatively high" is not justified, unless also other genes/proteins are analyzed for their dose response! Moreover, even with a 10-times less TGFbeta concentration, a robust downregulation is observed. This "extended time" experiment is puzzling, as already in Fig. 7A protein levels drop down to 25-30%
  12. Figure 7: Here the authors analyze isoform switching of RAC1 in response to TGFbeta and IIT. As the authors also refer to in the discussion, regulation of RAC1 alternative splicing was shown before and is regulated by ESRP1 and other RNA binding proteins. This is easy to test, by checking for their expression and performing siRNA mediated knockdown/overexpression experiments to prevent/induce isoform switching. The Rac1b overexpression can only demonstrate that Rac1b protein stability is not affected by TGFbeta.

Author Response

Dear Editor, dear Wendy:

This letter of submission is accompanied by our revised manuscript entitled:

“The quasimesenchymal pancreatic ductal epithelial cell line PANC-1 – a useful model to study clonal heterogeneity and EMT subtype shifting”

We are indebted to the reviewers for their enthusiastic and valuable comments and suggestions and have incorporated almost all of these into the revised version of our manuscript (highlighted in the “track changes” mode). We believe that the reviewers’ critiques have substantially improved the quality of our manuscript and we are looking forward to its final acceptance in Cancers.

Sincerely yours,

Hendrik Ungefroren

In the manuscript by Ungefroren and colleagues, the authors present the well-known and widely used pancreatic cancer cell line PANC1 as tool to analyze and experimentally modulate partial EMT states. Such tools are very valuable to address a very important research question about the function of epithelial-mesenchymal plasticity on tumor progression and metastasis. They used WB, qRT-PCR and invasion assays to characterize PANC1 heterogeneity. In addition, the response of derived single-cell clones and the parental cell line to TGFbeta and a cytokine cocktail of transdifferentiation (IFNγ, IL1β and TNFα) was analyzed. They found that single cell clones showed differential gene expression of some EMT markers and varying invasion capacities and TGFbeta response. Moreover, the parental line was used to further dissect changes in gene expression upon TGFbeta and IFNγ, IL1β and TNFα treatment, resulting in shifts towards more mesenchymal and epithelial phenotypes, respectively. A model that shows a partial EMT state under steady-state conditions and can be experimentally shifted towards both mesenchymal and epithelial states is of high value.

However, despite this importance, I miss a bit the novelty aspect of the findings to merit publication in “cancers”. For example, heterogeneity of Panc1 and other cell lines with respect to EMP was shown before, e.g. in (21).

Response: We are not aware of any published studies that have analysed the heterogeneity of the PANC-1 cell line, except the one that we cited (Ref. 23, Gradiz et al., 2016). Is it possible that this reviewer wanted to refer to another paper? In the paper by Aiello et al. (Ref. 21) heterogeneity of PANC-1 cells has not been studied. Rather, the authors attempted to group several established PDAC cell lines into those with complete (c) or partial (p) EMT and to compare them with the histologically defined quasimesenchymal (QM) or classical subtypes. Specifically, they used staining for membrane-associated ECAD (M-ECAD) to sort MIA PaCa-2 and PANC-1 cells predicted to have a c-EMT phenotype and others (i.e. BxPC-3) predicted to have a P-EMT phenotype and examined the sorted populations for Ecad mRNA and/or M-ECAD vs. intracellular (I-)ECAD. They found that MIA PaCa-2 and PANC-1 exhibited loss of Ecad mRNA and no intracellular ECAD protein upon loss of M-ECAD, while the PDAC cell lines exhibiting a p-EMT signature retained Ecad mRNA despite loss of M-ECAD, and a high percentage of M-ECAD-negative cells from these lines were positive for I-ECAD. They concluded that human tumor cells belonging to the QM subtype of PDAC utilize transcriptionally-dominated programs to lose their E phenotype during EMT while those belonging to the classical subtype rely on protein re-localization to lose their E phenotype during EMT.

The presented data are rather preliminary and the manuscript lacks a thread. To me it remains elusive whether the authors want to analyze the heterogeneity feature of PANC1 cells (in contrast to BxPC3) or to introduce a tool for EMP research. For the former part, the authors fail to correlate the clonal heterogeneity with specific features along the EMP/EMT axis.

Response: We thank the reviewer for bringing up this issue but actually we intended to do both analyze the clonal heterogeneity and introduce a cellular model for EMP research. We firmly believe that we have, in fact, correlated clonal variation with specific features of EMT/MET such as characterization with a bunch of EMT markers (Fig. 1 and 3), migratory activity (Fig. 2), response to TGFb1 (Fig. 3) and transdifferentiation potential/cellular plasticity (Fig. 4). Moreover, in response to requests from Reviewer 1, we have identified strong differences among clones, not only with respect to the ECAD:VIM ratio but also the relative ratio of RAC1b (epithelial marker):RAC1 (mesenchymal marker), and, in addition, have now shown that the different EMT phenotypes of the clones remain stable during extended time in culture. Please, see new Figures S2, S4B and 1C for details.   

Most experiments either need some more in depth analysis or describe observations that are not new. More specific points are listed below:

  1. In general, the choice of EMT/EMP markers are chosen a bit random (like SYP, RAC1, Snail, Serpine, Claudin4 (but not other EMT-TFs). Why is SYP only analyzed on transcript level? More markers need to be assessed to characterize the clone identity in respect to the EMP continuum (e.g. Snail/E-cad expression levels in steady-state, TGFbeta response, IIT response, migration/invasion: are these features correlated?)

Response: It may well be that we have not made it clear enough why we have chosen these markers. SYP was not meant to be an EMT marker but a prototype marker of NE differentiation. Its protein expression in PANC-1 cells was too low to be able to safely identify quantitative differences among individual clones. For this reason, we decided to detect it at the mRNA level, which revealed strong differences between individual clones. We have chosen classic EMT markers like ECAD, VIM, SNAIL1 (see immunoblots in Fig.1 and 3C, respectively) and - in response to a request of Reviewer 1 - also RAC1b (see new figures S2 and S4B). In response to the above comment (“but not other EMT-TFs”), we have performed qPCRs for SNAIL2/ SLUG (now included as new panel D in Figure 3). PAI-1 and CLDN-4 were primarily chosen as an established and highly responsive TGFb target gene and metastasis-associated gene in PDAC, respectively, rather than EMT markers. The relative ECAD:VIM ratio in steady-state/under basal conditions is presented in Figure 1A and reveal strong differences among clones! The SNAIL/ECAD protein and SLUG mRNA expression levels in steady-state/basal conditions are presented in Figure 3C and D, respectively. For the TGFb1 response, we have analysed now four targets genes, SERPINE1 (Fig. 3B) and SLUG (new Fig. 3D) at the mRNA level as well as SNAI1 and CDH1 at the protein level (Figure 3C), which we believe should suffice. The IIT response was analysed on two genes (INS and NEUROG3) in contrast to most other studies, in which either one of both has been measured (e.g. Ref. 30 in the revised ms.). The issue of correlation was also brought up by Reviewer 1 (please see his points 2, 4 and 5 and my responses to them). As shown for ECAD and VIM in Fig. 1A, we have calculated the relative RAC1b:RAC1 protein expression ratios and found them to correlate with the respective migratory activities (see new Figures S4A (for ECAD:VIM ratios) and Fig. S4B (for RAC1b:RAC1 ratios)). There also seems to be a correlation between EMT phenotype and the TGFb1 response (see my response to point 4 of Reviewer 1).

  1. The order of the clones and in general the order of WB (e.g. Fig. 6: Rac1 top, Vim bottom (A) and vice versa (B) is changing multiple times, which makes it hard to compare

Response: We totally agree with the reviewer on this point and regret if this has caused confusion. We have changed the order of the RAC1 and VIM blots in Figs. 6A and B. However, the order of the clones cannot be changed so easily as this would require cutting and cropping of blots or redoing of all the Westernblots. We would therefore prefer to leave the order of the clones in the WBs as it stands.

  1. Figure S1: Phase contrast image of the parental cell line should be shown as well

Response: A phase contrast image of parental PANC-1 cells (untreated control to TGFb1-treated cells) is shown in Figure S8 (formerly Figure S4). A short sentence has been added to the legend of Fig. S1A to indicate this.

  1. 2: parental PANC1 and BxPC3 should be included

Response: As requested, the migration curves of parental PANC-1 and BxPC3 cells have been included as Figure S5 in relation to two clones each shown in Figure 2A and 2B.

  1. 3A etc.: is fold-change or percentage shown here?

Response: We apologize for the lack of clarity. Percent of control (set arbitrarily at 100) is shown here. In the revised version, this has been specified in the figure legend.

  1. Figure 3: The authors aim on analyzing proliferation. However, without assessing cell death, the data only show relative cell numbers over time, but no indication about proliferative activities. What about the differences in cell numbers/proliferation under steady-state conditions? This aspect is very important otherwise relative changes do not mean much. Moreover, since BxPC3 clones did not show EMP differences, it would be interesting to see their performance here as well.

Response: The data in Figures 3A and the former Figure S2 (now Figure S6B) were meant to determine specifically the effect of TGFb1 on cell viability and proliferation, respectively. It should be stressed that although PANC-1 cells have remained responsive to this growth factor, TGFb1 does not induce apoptosis in these cells under the conditions applied (standard growth medium with 10% fetal calf serum, TGFb1 concentration of 5 ng/ml, and treatment periods of 3-6 days). In contrast, through induction of EMT, TGFb can even decrease apoptosis by inducing chemoresistance (PMID: 23300530). Accordingly, the number of dead cells in all our assays was very low (< 1% as determined by trypan blue exclusion assay) in all clones regardless of TGFb1 treatment, indicating that apoptosis cannot account for the observed differences in cell counts. This is in agreement with several of our earlier studies on the cytostatic effect of TGFb1 on the parental cells (i.e., PMID: 31108998, PMID: 21624123, PMID: 21225226). With respect to differences in cell numbers/proliferation under steady-state conditions we thank the reviewer for this question and have performed cell counting assays with PANC-1 clones in steady-state/basal conditions. We observed clear differences in growth after 72 hours of incubation in standard growth medium. These data have been included in Figure S6 (formerly S2) as new panel A in the revised version (the former graph in Figure S2 became Figure S6B). We agree with this reviewer that it would be nice to also evaluate basal proliferative activities for the BxPC3 clones, however, since we focus here on PANC-1 cells and growth regulation is not immediately relevant for EMP, we would prefer not to include these data in this manuscript.

  1. Figure 4B: I am puzzled by the rational of this experiment. The results are completely different to the original observation by (30), maybe because of highly variable plasmid DNA uptake per cell by transient transfection in comparison to transduction. In any case, I do not see why these experiments are crucial for identifying clonal variation and describing EMP differences. Shouldn't these graphs also include a measurement without any synchronization (ctrl)? What is the conclusion of this experiment with regard to EMP?

Response: We would like to clarify that our goal was to evaluate clonal heterogeneity/variation in molecular clock function rather than associating this function with different EMP phenotypes. This is now clearly stated at the end of section 2.4. (“establishing a relationship between clock gene function and EMT phenotype was not possible”). The differences between our findings and those of Li et al., 2020 (originally Ref. 30) may have been caused by the method of gene transfer - lentiviral transduction vs. transient transfection. As suggested by the reviewer, transient transfection may be associated with highly variable plasmid DNA uptake per cell in comparison to transduction. Other parameters that may have contributed to this discrepancy is the heterogeneity of this cell line itself, different culture conditions, and the fact that Li and coworkers synchronized the cells with a medium change, which is a weaker synchronizing strategy than the different synchronization methods we have been using. It should be stressed that even if a variability in transfection efficiency occurred, it cannot explain the absence of circadian responses in some clones, since baseline raw luminescence signals were comparable across all clones. Without synchronisation the individual circadian rhythms of the cells will cancel each other out so that at the “well-level” no rhythms will be observed anymore. For this reason, a synchronisation event is always required.

  1. Figs 5-7: After characterization of the single cell clones, why do the authors switch back to the parental cell line?

Response: This is a reasonable question. However, the second part of our manuscript was designed to demonstrate phenotype shifting of the parental cells (which has now been exclusively stated in the subheading and the 3rd sentence of section 2.5. to avoid any misunderstanding). Demonstrating all the events for each of the six clones would have gone beyond the scope of this study. Since we have shown in Figure 3 that all clones respond - albeit to a different extent - to TGFb1, we believe that it would suffice here to study the behavior of the parental cells. For this reason, we would prefer not to repeat the experiments shown in Figs. 5-7 with single clones, although we agree with the reviewer that analyzing the single clones would surely provide additional information. However, the focus here is on the identification of agents that are able to induce phenotype shifting rather than on clonal heterogeneity.

  1. Figure 6: The IIT treatment should also be done for the single cell clones! To me, the important question is, are Panc1 cells representing a heterogeneity that is reflected in the clonal expansion, do the clones all respond in a similar/different way to the various treatments, like the parental cells, but with varying extend? Do these clones represent specific states within the EMP continuum?

Response: Please see point 10 for a justification of why we have proceeded with the parental cells rather than the single clones. In fact, we have done the IIT treatment of six single clones in Figure 4A to study its effects on INS and NEUROG3 expression. In future studies we also plan to analyse IIT-treated single clones for expression of all the E and M markers used in Fig. 6 for characterization of the parental cultures. The clones all responded in a similar way to these treatments but to a varying extent. Based on the phenotypic and functional heterogeneity shown in Figures 2-4 and the known responsiveness to EMT and MET-inducing agents we believe that the clones present in parental cultures represent specific states in the EMP continuum. However, to further characterize these states, more specific markers need to be applied to our clones such as those identified in Ref. 42 of the revised manuscript. We also believe that repeating all the experiments shown in Figure 7 with single clones would not provide much additional information on the use of RAC1 and RAC1b as markers of phenotype switching.

  1. Figure 6D: The data of CK19 and E-cad should be shown even if there is no difference.

Response: We agree with the reviewer and have added the CK19 data for MIA PaCa-2 to Fig. 6D. However, addition of another graph with the ECAD data of PANC-1 would destroy the layout of the figure and we therefore prefer to mention these data as “data not shown”.

  1. Figure 7B: what do the authors conclude from the ALK5/MAPK inhibitor treatment? Is non-canonical TGFbeta signaling responsible for alternative RAC1 splicing? The authors state that “relatively high” TGFbeta doses are necessary for a robust RAC1b decrease. However, 5 ng/ml were used throughout the paper and is also used by many others. So, the term "relatively high" is not justified, unless also other genes/proteins are analyzed for their dose response! Moreover, even with a 10-times less TGFbeta concentration, a robust downregulation is observed. This "extended time" experiment is puzzling, as already in Fig. 7A protein levels drop down to 25-30%

Response: We thank the reviewer for pointing out the lack of clarity. The use of the ALK5 and MAPK inhibitors is only to show that the TGFb1-induced decrease in RAC1b protein levels is ALK5- but not p38 MAPK-dependent. Hence, this effect does not involve p38 MAPK and is likely mediated by canonical Smad signaling (a possible involvement of ERK and JNK, however, may still be possible). Another reason for using SB203580 was the fact that it is chemically/structurally very similar to SB431542 and thus served as a “negative control” for SB431542. Regarding the use of the term “relatively high” we perfectly agree with this reviewer and have thus removed it. Moreover, we have rephrased the sentence to also remove the statement on the extended treatment time.

  1. Figure 7: Here the authors analyze isoform switching of RAC1 in response to TGFbeta and IIT. As the authors also refer to in the discussion, regulation of RAC1 alternative splicing was shown before and is regulated by ESRP1 and other RNA binding proteins. This is easy to test, by checking for their expression and performing siRNA mediated knockdown/overexpression experiments to prevent/induce isoform switching. The Rac1b overexpression can only demonstrate that Rac1b protein stability is not affected by TGFbeta.

Response: Regulation of RAC1b alternative splicing and its regulation by ESPR1 has indeed been described before. However, neither TGFb1 nor IIT (or any of these three cytokines) has yet been identified as an inhibitor or activator, respectively, of RAC1b expression! Our findings are therefore totally novel. Although clearly important, we feel that elucidating the molecular mechanism of isoform switching and in particular the role of ESPR1 in this process is clearly beyond the scope of our manuscript, but is nevertheless important and may be done in future studies (this statement has been added to this paragraph). Rather, our intention was to identify RAC1b and RAC1 levels, or their ratio, as biomarker(s) to monitor mesenchymal conversion, i.e., by TGFb1, or epithelial conversion, i.e., by IIT. Regarding the conclusions made from Figure 7C, we have added that ectopic Rac1b (over)expression only demonstrates that Rac1b protein stability is not affected by TGFb1, suggesting the reverse conclusion that the inhibitory effect of TGFb1 on endogenous RAC1b involves a decrease in transcription rather than in protein stability.

Additional changes made

  1. Due to the addition of various new Supplementary figures, the numbering of the original Suppl. figures has been updated as follows:

Figure S1A: Morphology and growth patterns of single cell-derived clones of PANC-1.

Figure S1B (new): Morphology and growth patterns of single cell-derived subclones of PANC-1 clones P3D2 and P4B11.

Figure S2 (new): Quantification of RAC1 and RAC1b protein levels in PANC-1 single clones.

Figure S3 (formerly Fig. 1C): Expression analysis of the neuroendocrine marker SYP in the indicated PANC-1 clones.

Figure S4 (new): Correlation of the ECAD/VIM and RAC1b/RAC1 ratios with migratory activities in six individual PANC-1 clones.

Figure S5 (new): Migratory activity of parental PANC-1 or BxPC-3 cells compared to two single cell-derived clones.

Figure S6 (in part new, formerly S2): Proliferative responses of individual PANC-1 clones under basal conditions and after stimulation with TGFβ1 as determined by cell counting assay.

Figure S7 (formerly S3): Rhythmic parameters evaluation by CircaSingle.

Figure S8 (formerly S4): Morphology and growth pattern of parental PANC-1 cultures treated with TGFβ1.

Figure S9 (formerly S5): Immunoblot analysis of SNAIL, RAC1 and VIM in parental PANC-1 cells after treatment with IIT.

S10-S14: Uncropped blots.

  1. For the sake of consistency, the terms “epithelial” and “mesenchymal” have been replaced by their abbreviations “E” and “M”, respectively.
  2. Two new references have been added to the References section.
  3. In Figure 1A and 1C, the labeling of the ordinates has been changed from “Ratio ECAD/VIM” to “Relative ratio ECAD:VIM” since the ECAD and VIM signals were detected independently with different exposure times.

Round 2

Reviewer 1 Report

Figure S4: A. The order of the clones should be sorted according to the Ratio ECAD/VIM. Then it’s easier to see that clones with low ECAD/VIM ratio have high migratory activity. The same for fig S4B.

Otherwise, the authors sufficiently revised the manuscript, which is in my opinion now suitable for publication.

Reviewer 3 Report

The authors addressed all my points appropriately.